# Cortical excitability controls the strength of mental imagery

**Rebecca Keogh[1][†]\***, **Johanna Bergmann[1,2,3][†]**, **Joel Pearson[1]**

[1]School of Psychology, University of New South Wales, Sydney, Australia;
[2]Department of Neurophysiology, Max Planck Institute for Brain Research, Frankfurt, Germany; [3]Brain Imaging Center Frankfurt, Goethe-University Frankfurt, Frankfurt, Germany

**Abstract** Mental imagery provides an essential simulation tool for remembering the past and planning the future, with its strength affecting both cognition and mental health. Research suggests that neural activity spanning prefrontal, parietal, temporal, and visual areas supports the generation of mental images. Exactly how this network controls the strength of visual imagery remains unknown. Here, brain imaging and transcranial magnetic phosphene data show that lower resting activity and excitability levels in early visual cortex (V1-V3) predict stronger sensory imagery. Further, electrically decreasing visual cortex excitability using tDCS increases imagery strength, demonstrating a causative role of visual cortex excitability in controlling visual imagery. Together, these data suggest a neurophysiological mechanism of cortical excitability involved in controlling the strength of mental images.

## Introduction

Visual imagery - the ability to 'see with the mind's eye' - is ubiquitous in daily life for many people; however, the strength and vividness with which people are able to imagine varies substantially from one individual to another. Due to its highly personal nature, the study of visual imagery has historically relied on self-report measures and had long been relegated to the shadows of scientific inquiry. However, with the advent of fMRI and new analysis techniques like decoding, as well as new advances in behavioral and psychophysical experiments, this is quickly changing (**Pearson, 2014**; **Pearson, 2019**).

To date, much of the research in the field of visual imagery has focused on the similarities between visual imagery and perception, due to a long-ranging debate around whether visual imagery can be depictive and/or pictorial, referred to as the 'imagery debate' (**Pearson and Kosslyn, 2015a**). Research has shown that a large network of occipital, parietal, and frontal areas are involved when imagining (**Pearson et al., 2015b**; **Dijkstra et al., 2019**), with recent studies providing evidence that visual imagery content is tied to early visual cortex, indicating that imagery-related processing overlaps with that of perception. For example, research using fMRI has demonstrated that BOLD activity in early visual cortex increases when individuals imagine, and the content of visual imagery can be decoded from early visual cortex, as well as being cross-decoded from perception (**Albers et al., 2013**; **Thirion et al., 2006**; **Cui et al., 2007**). Additionally, recent work has shown that trial-by-trial self-rated vividness of visual imagery during an imagery task correlated with the neural overlap between perception and imagery (**Dijkstra et al., 2017**). Brain stimulation research has similarly investigated whether the early visual cortex is involved during visual imagery with findings demonstrating that, like motor imagery, visual cortex excitability increases during imagery (**Cattaneo et al., 2011**; **Sparing et al., 2002**).

It is now well accepted that visual imagery can indeed be pictorial/depictive in nature and involves representations in low-level visual cortex (**Pearson and Kosslyn, 2015a**). However, there

**\*For correspondence:**
rebeccalkeogh@gmail.com

[†]These authors contributed equally to this work

**Competing interests:** The authors declare that no competing interests exist.

are also large individual differences in the reported vividness of imagery across the general population. Some report imagery so vivid it is akin to seeing the image, while others report no experience of visual imagery at all, a new special population referred to as aphantasia (*Zeman et al., 2015*). These large individual differences exist in both subjective reports (*Galton, 1883*), and in objective measures of imagery strength (*Keogh and Pearson, 2018*). Little research has investigated exactly what drives these large individual differences. One study reported that the vividness of visual imagery correlates positively with BOLD activity changes in visual cortex during an imagery task (*Cui et al., 2007*). Another study found a correlation between imagery vividness and the similarity of BOLD responses for perception and imagery in early visual cortex (*Lee et al., 2012*). A recent study found that trial-by-trial differences in imagery vividness were also related to the similarity of BOLD responses between imagery and perception (*Dijkstra et al., 2017*).

Taken together, these studies suggest that visual cortex is linked to the subjective vividness of visual imagery. However, they do not provide information about why some individuals are better at recruiting the early visual cortex to create stronger more vivid images. Work in synesthesia and migraines has found evidence that the neural excitability of early visual cortex relates to the experience of *involuntary* forms of visual imagery (*Terhune et al., 2015a*; *Terhune et al., 2011*; *Gunaydin et al., 2006*). Specifically, these previous studies have shown that individuals who experience grapheme-colour synesthesia, or auras prior to the onset of migraines, have heightened visual cortical excitability measured by TMS phosphene thresholds (*Terhune et al., 2015a*; *Terhune et al., 2011*; *Gunaydin et al., 2006*). It is known that the excitability of visual cortex varies substantially across individuals, and as such may be a candidate for driving some of the observed interindividual differences in visual imagery strength.

Here, we investigated whether cortical excitability might also be linked to the individual differences that exist in the strength of voluntarily produced visual imagery. We used a multi-method approach (fMRI, TMS, and tDCS, see Materials and methods for measures of cortical excitability) to assess the potential contributions of resting levels of cortical excitability in the visual imagery network as a critical physiological precondition, which influences the strength of visual imagery.

## Measuring visual imagery strength

To measure mental imagery strength, we utilized the binocular rivalry imagery paradigm (see *Figure 1*), which has been shown to reliably measure the sensory strength of mental imagery through its impact on subsequent binocular rivalry perception (*Pearson, 2014*). Previous work has demonstrated that when someone imagines a pattern or is shown a weak perceptual version of a pattern, they are more likely to see that image in a subsequent brief binocular rivalry display (see *Pearson et al., 2015b* for review of methods). Longer periods of imagery generation, or weak perceptual presentation, increase the probability of perceptual priming of subsequent rivalry. For this reason, the degree of imagery priming has been taken as a measure of the sensory strength of mental imagery. Importantly, this measure of imagery is directly sensory; while it is related to subjective reports of imagery vividness, it is not a direct proxy for subjective reports of imagery vividness, and findings regarding their relationship across individuals have been mixed (see *Figure 1—figure supplement 1A* and *Pearson et al., 2011*; *Bergmann et al., 2016a*). This measure of imagery strength has been shown to be both retinotopic location and spatial orientation specific (*Bergmann et al., 2016a*; *Pearson et al., 2008a*), is reliable when assessed over days or weeks (see *Figure 1—figure supplement 2* and *Bergmann et al., 2016a*), is contingent on the imagery generation period (therefore not due to any rivalry control) and can be dissociated from visual attention (*Pearson et al., 2008a*). This measure of imagery is advantageous in that it allows us to avoid the prior limitations of subjective introspections and reports.

## Results

### Visual cortex and visual imagery strength

Correlations between visual cortex excitability and visual imagery strength: exploratory fMRI analysis

First an exploratory analysis was run to see if there was any relationship between cortex physiology and imagery strength. To do this, we looked at fMRI data and assessed a sample of 31 participants

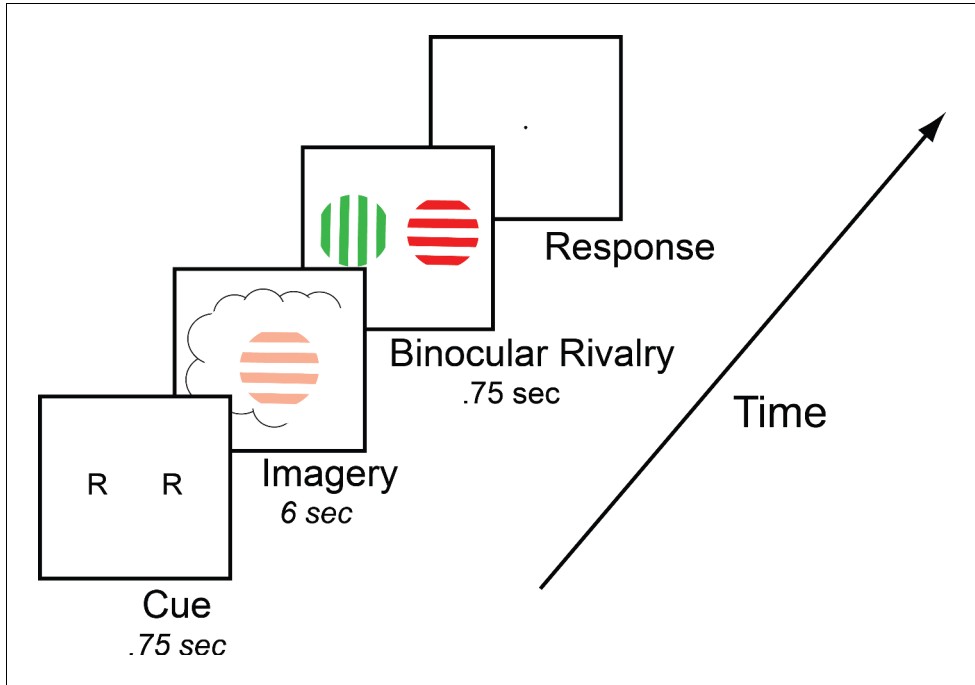

**Figure 1.** Timeline of the basic imagery experiment. Participants were cued to imagine a red-horizontal or a green-vertical Gabor patch for 6–7 s by the letter R or G (respectively). Following this, they were presented with a brief binocular rivalry display (750 ms) and asked to indicate which image was dominant. In the behavioral experiments with the brain-imaging sample and in three of the tDCS experiments, a rating of subjective vividness of the imagery also preceded the binocular rivalry display.

The online version of this article includes the following figure supplement(s) for figure 1:

**Figure supplement 1.** Imagery vividness results.

**Figure supplement 2.** Re-test reliability for imagery strength (A) and Phosphene Thresholds (B).

resting-state fMRI data (these participants form part of a sample that has previously been reported on in *Bergmann et al., 2016a*; *Bergmann et al., 2016b*); however, these previous analyses were structural rather than functional). We related this data set to each individual's imagery strength determined using the binocular rivalry method (% primed, see *Figure 1*). Using a whole-brain surface-based group analysis (see Methods), we found that the normalized mean fMRI intensity of clusters in the visual cortex showed a negative relationship with imagery strength, while frontal cortex clusters showed positive relationships (multiple comparison-corrected; see *Figure 2*, left column; and *Supplementary file 1* Supplementary Table S1 –S3). This pattern of results was also present in a second set of resting-state fMRI data that was acquired from the sample (see *Figure 2*, right column). We also assessed the relationship between retinotopically defined early visual cortices V1-V3 and the adjacent occipito-parietal areas (defined by the Desikan–Killiany atlas). We found significant negative correlations with normalized mean fMRI intensity levels in V1-V3 and lateral occipital cortex and imagery strength (See *Figure 2—figure supplement 1*).

## Correlations between visual cortex excitability and visual imagery strength: TMS

To further substantiate our observations and circumvent other potential confounds that might influence the fMRI data (see appendices for discussion of these limitations), we next utilized a different methodology that measures cortical excitability: transcranial magnetically induced phosphenes. A new sample of 32 participants performed an automated phosphene threshold (PT) procedure using transcranial magnetic stimulation (TMS) over early visual cortex. Visual phosphenes are weak hallucinations caused by TMS applied to visual cortex. The magnetic strength needed to induce a phosphene is a reliable and non-invasive method to measure cortical excitability (see Materials and

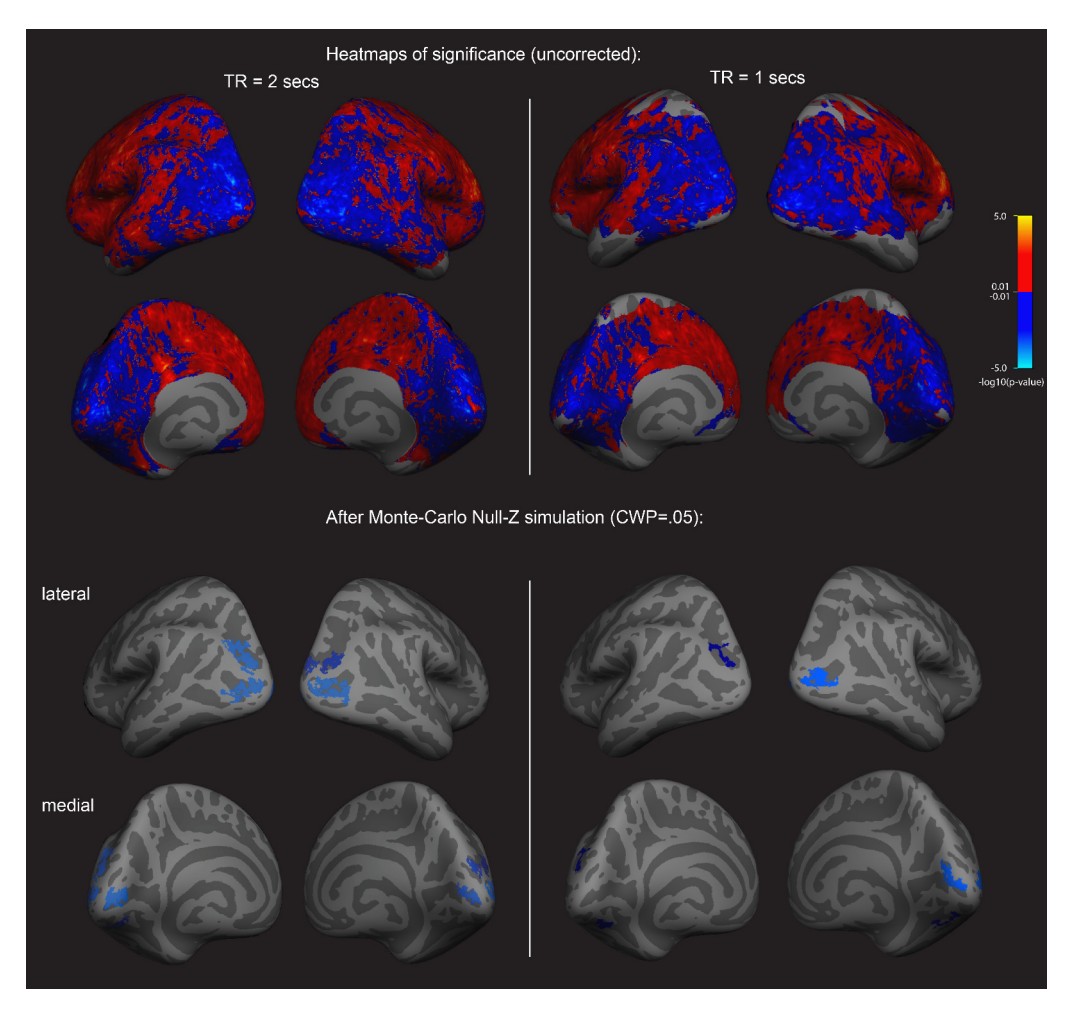

**Figure 2.** Surface-based whole brain analysis of data from two different fMRI resting-state measurements: negative associations with imagery strength in the occipital cortex. Two columns on the left: results of the main resting-state fMRI data set with a TR of 2 s (TR2). Two columns on the right: results of an additional resting-state fMRI data set with a TR of 1 s (TR1); in those participants with which both measurements were conducted, about half were done on the same day. In the other half, the two measurements were conducted on different days. The two upper rows show the uncorrected (positive and negative) relationships with imagery as heatmaps. The two lower rows show the corrected clusters that had a *negative* association with individual imagery strength at a cluster-wise probability threshold (CWP) of p<0.05 (also see *Supplementary file 1 - Supplementary Table S1*). The two hemispheres are shown from the back, with the lateral view in the upper and the medial view in the lower panel. Multiple comparison correction was done using Monte Carlo Null-Z simulation (mc-z). No smoothing of the functional mean intensity data was applied. In line with the correlation analyses using normalised fMRI mean intensity of atlas- and retinotopically defined areas, only fMRI mean intensity clusters in the back of the brain, where early visual and lateral occipital cortex are located, showed negative associations with imagery strength (% primed). The fMRI measurement with a TR = 2 s has a better signal-to-noise ratio, as longer TR increase T2* tissue contrast (e.g. see *Hashemi et al., 2010*); in addition, the larger voxel size of the TR1 measurement (3.28 × 3.28×5 mm$^3$) also means that they are more likely to pick up signals from other tissue (e.g. white matter), thereby increasing the contributions of biophysical noise. Both of this likely weakens the observed correlations with behavior; this might explain why none of the relationships with the brain areas using retinotopic mapping and the Desikan-Killiany atlas survived multiple comparison correction in the ROI-based approach (all p>0.05). Despite this, the clusters from the two different measurements in the surface-based group analysis show striking similarities; while the clusters in the TR1 measurement are smaller and sparser, their location in early visual and lateral occipital cortex are strongly overlapping with those found in the TR2 measurement. Further analyses showed that these similarities were not driven by the group that completed the measurements on the same day (analysis not shown).

The online version of this article includes the following source data and figure supplement(s) for figure 2:

**Figure supplement 1.** Retinotopic ROI anslysis of resting-state fMRI data and it's realtionship with imagery strength.

**Figure supplement 1—source data 1.** fMRI resting state correlation data.

**Figure supplement 2.** Surface-based brain analysis of data from two different fMRI resting-state measurements and imagery: positive associations with imagery strength in the frontal cortex.

**Figure supplement 3.** Visual cortex relationships with imagery strength and the number of EPI volumes discarded at the beginning of the run.

methods section for explanation of phosphene thresholds and cortex excitability). In line with the normalized mean fMRI intensity data, we found a significant negative correlation between imagery strength and visual cortex excitability (data shows inverse phosphene threshold (100-PT) for ease of visualizing data as PT's are negatively correlated with cortical excitability: $r = -0.43$, p=0.0127; *Figure 3A*). In other words, individuals with lower visual cortex excitability exhibited stronger imagery. Importantly, we also tested the phosphene threshold retest reliability for our paradigm over 2 days and found it was a very reliable measure ($r_s = 0.75$, p<0.001; see *Figure 1—figure supplement 2*), as was our measure of imagery strength re-test reliability of tDCS experiments imagery strength: $r_s = 0.51$, p<0.001; see *Figure 1—figure supplement 2*).

To assess possible effects of a decisional bias, mock rivalry trials were included in all tests of imagery strength (*Pearson et al., 2008a*; *Bergmann et al., 2016a*; *Keogh and Pearson, 2011*; *Keogh and Pearson, 2014*)(see Materials and methods). We found no correlation between real binocular rivalry and 'mock priming' (fMRI (circles $r_s = -0.03$, p=0.89 and TMS (squares) $r_s = -0.01$, p=0.97, see *Figure 3B*). These data suggest it is unlikely that the relationship between imagery strength and physiology is due to demand characteristics or decisional bias.

## Manipulating visual cortex excitability using tDCS

The data suggest that the excitability of the visual cortex might influence the strength of visual imagery, as participants with lower visual cortex activity tended to have stronger visual imagery and vice versa. However, these data do not speak to the causal role of early visual cortex in creating strong mental images. If the association between imagery strength and visual cortex activity is causal, manipulating visual cortex excitability should likewise modulate imagery strength.

To assess this hypothesis, we utilized non-invasive transcranial direct current stimulation (tDCS), which can increase or decrease cortical excitability depending on electrode polarity and position (see *Filmer et al., 2014* for review, and methods for evidence of tDCS modulating visual cortex excitability). Broadly speaking, when the cathode is placed over the cortex, when averaging across participants, the underlying cortical excitability is decreased, whereas the anode increases excitability (although the polarity specific effects can be influenced by multiple methodological and inter-individual differences *Strube et al., 2016*; *Filmer et al., 2019a*; *Batsikadze et al., 2013*; *Monte-Silva et al., 2013*). Sixteen new participants underwent both anodal and cathodal stimulations of visual cortex (see *Figure 4B* for electrode montage) on 2 separate days (separated by at least 24 hr).

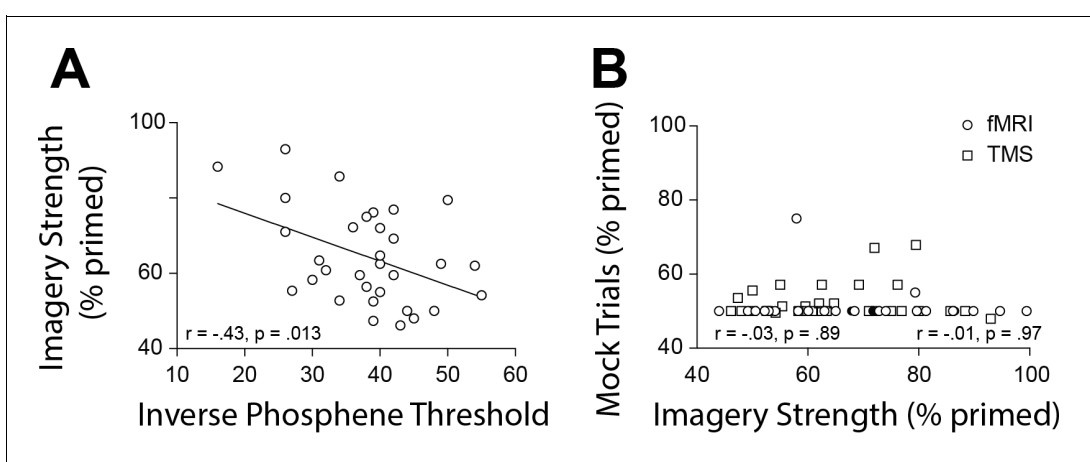

**Figure 3.** Scatterplots for TMS phosphene thresholds and mock rivalry data. (**A**) Correlation between the inverse phosphene threshold and imagery strength. Individuals with lower cortical excitability in visual cortex tended to have stronger imagery. (**B**) Correlation between mock priming scores and real binocular rivalry priming for participants in the fMRI (circles) and TMS (squares) study. There was no significant association between perceptual priming in real and mock trials for the fMRI or TMS data. In the scatterplots (A & B), each data point indicates the value of one participant; the bivariate correlation coefficients are included with their respective significance levels. The online version of this article includes the following source data for figure 3:

**Source data 1.** TMS inverse phosphene correlation data.

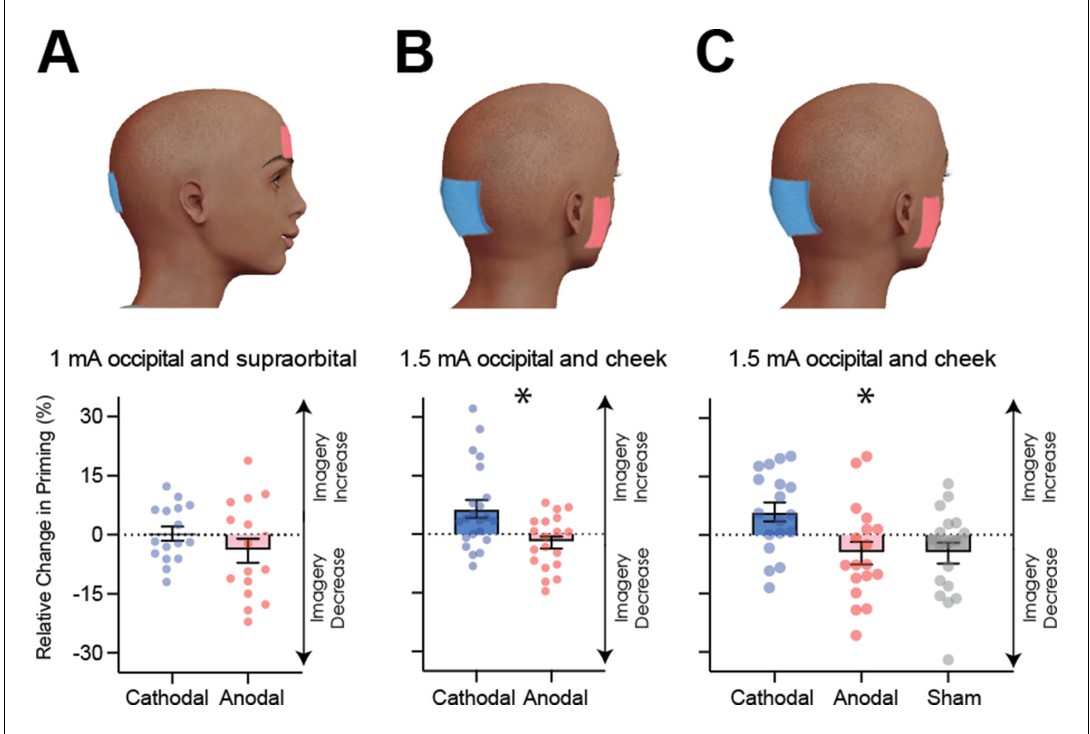

**Figure 4.** Visual cortex stimulation data. (**A**) Effect of visual cortex stimulation on imagery strength at 1mA. The top image shows the tDCS montage, with the active electrode over Oz and the reference electrode on the supraorbital area. The bottom image shows the effect of cathodal (decreases excitability, blue dots represent each participant's data) and anodal (increases excitability, red dots represent each individual participant's data) stimulation averaged across all tDCS stimulation blocks (D1, D2, P1, and P2). (**B**) Effect of visual cortex stimulation on imagery strength at 1.5mA. Top: the tDCS montage with the active electrode over Oz and the reference electrode on the right cheek. Bottom: the effect of cathodal (blue dots, decrease excitability) and anodal (red dots, increase excitability) stimulation averaged across all blocks during and after tDCS stimulation (D1, D2, P1, and P2). Each data point represents a single participant. Imagery strength increases in the cathodal stimulation condition (blue), when neural excitability is reduced. (**C**) Effect of visual cortex stimulation on imagery strength at 1.5mA. Top: the tDCS montage with the active electrode over Oz and the reference electrode on the right cheek. Bottom: The left bar shows the relative change in imagery strength for cathodal stimulation (blue bar, blue dots represent individual participants data), the middle bar shows the relative change in imagery strength for anodal stimulation (red bar, red dots represent individual participants data), while the right bar shows the change in imagery strength for sham stimulation (grey bar, grey dots represent individual participants data). All error bars show ± SEMs and stars (*) indicate a significant effect of tDCS polarity.

The online version of this article includes the following source data and figure supplement(s) for figure 4:

**Source data 1.** 1mA occipital tDCS data.
**Source data 2.** 1.5mA occipital tDCS data.
**Source data 3.** 1.5mA occipital and sham tDCS data.
**Source data 4.** 1.5mA occipital TMS + tDCS data.
**Figure supplement 1.** Raw tDCS imagery strength and difference scores as a function of block for experiments 1, 2 and 4.
**Figure supplement 2.** Raw tDCS imagery strength and difference scores as a function of block for experiments 3 and 5.
**Figure supplement 3.** tDCS modulation of phosphene thresholds.

On each day, participants completed six blocks of the imagery task, two before tDCS, two during tDCS and two post-tDCS (see *Figure 4—figure supplement 1A* for experimental timeline). To assess the effect of tDCS on imagery strength, we calculated the percent change in priming for each participant from baseline (on each day, see Materials and methods for percent change calculation details) such that positive numbers indicate increases in imagery strength and negative ones indicate decreases.

*Figure 4A* shows relative imagery priming percent change scores averaged across all stimulation blocks with 1mA of tDCS stimulation (data per block can be seen in *Figure 4—figure supplement 1C*). Linear mixed-effects analysis were computed for all following tDCS experiments. This analysis was run with a 2 (tDCS polarity: cathodal and anodal), x 4 (block: D1, D2, P1, P2 – see *Figure 4—figure supplement 1A* for timeline and *Figure 4—figure supplement 1C* for data for each block) x 2

(order of stimulation: cathodal on the first or second day) design. When fitting a linear mixed model, the effect of tDCS polarity was not significant ($\chi^2(1)$=2.99, p=0.084).

The non-significant results from the first tDCS experiment may be due to the stimulation intensity of 1 mA being too low to produce any effect - many tDCS studies use an intensity ranging from 1.5-2mA (for example see *Jacobson et al., 2012*). To investigate whether the lack of a significant result with 1mA was due to the low stimulation intensity, we ran a second tDCS study with a higher intensity of 1.5mA (see Materials and methods) and both cathodal (blue bars and dots) and anodal (red bars and dots) stimulation conditions. Additionally, to ensure we were not also stimulating the prefrontal cortex, the supraorbital placement of the reference electrode was moved to the cheek (*Figure 4B*). A linear mixed-effects analysis was run with a 2 (tDCS polarity: cathodal and anodal), x 4 (block: D1, D2, P1, P2 – see *Figure 4—figure supplement 1A* for timeline and *Figure 4—figure supplement 1E* for data for each block) x 2 (order of stimulation: cathodal on the first or second day) design. The effect of tDCS polarity was significant $\chi^2(1)$=15.85, p=6.86e$^{-05}$. The changes were in line with the correlational data for resting levels of visual cortex excitability and activity (see *Figures 2* and *3*), such that imagery strength increased when visual cortex excitability was decreased (cathodal stimulation, see *Figure 4B*), while the opposite was true of increasing visual cortex excitability (anodal stimulation).

It is likely that the change from 1mA to 1.5mA allowed us to observe the modulatory effects of tDCS; however, it also might be that the change in montage had an influence (i.e. location of reference electrode). Further, it may be the case that there are either fatigue or practice effects on this visual imagery task, that is perhaps participants just get better/worse on this task due to doing multiple sessions. For this reason, a third experiment was run to assess the effects of fatigue/practice and the change of reference location. This study was identical to the above study with the inclusion of a sham condition where the tDCS machine shut off after 30 s of stimulation. A linear mixed-effects analysis was run with a 3 (tDCS polarity: cathodal, anodal and sham), x 4 (block: D1, D2, P1, P2 – see *Figure 4—figure supplement 2A* for timeline and *Figure 4—figure supplement 2C* for data for each block) x 3 (order of stimulation: cathodal on first, second or third day) design. The effect of tDCS polarity was again significant $\chi^2(2)$=21.66, p=1.98e$^{-05}$. These data indicate that cathodal stimulation results in increased imagery strength (see *Figure 4C*), and this is unlikely to be a practice effect, as sham stimulation results in decreases in imagery strength. Additionally, previous work using the same binocular rivalry paradigm has demonstrated no increases in visual imagery strength after multiple days of training (*Pearson et al., 2011*). Taken together, these data suggest that cathodal stimulation leads to increases in imagery strength due to decreased visual cortex excitability, and these changes cannot be explained as a learning effect due to performing multiple sessions of the imagery task.

Although other studies have provided evidence that tDCS does change the excitability of the visual cortex (see *Antal et al., 2003* for example), we wanted to ensure that our specific stimulation paradigm was indeed modulating visual cortex excitability. We ran a separate control study comparing TMS-phosphene thresholds before and after the same tDCS paradigm (1.5mA, active electrode on Oz and reference on the cheek, see *Figure 4B*, all subjects received both anodal and cathodal stimulations across separate days; see Materials and methods for further details). If our cathodal stimulation is decreasing visual cortex excitability, greater TMS power output would be required to elicit phosphenes post-cathodal stimulation, whereas post-anodal stimulation we would predict the opposite effect. A linear mixed-effects analysis was run with a 2 (tDCS polarity: cathodal and anodal), x 2 (block: Pre tDCS and Post tDCS) x 2 (order of stimulation: cathodal on the first or second day) design. We found that phosphene thresholds measured immediately after anodal stimulation decreased, whereas after cathodal stimulation phosphene thresholds increased (significant effect of tDCS polarity ($\chi^2(1)$=4.32, p=0.038, see *Figure 4—figure supplement 3*)). These findings show that our stimulation paradigm changes cortical excitability in the expected direction, that is cathodal stimulation decreases cortical excitability, whereas anodal stimulation increases activity.

## Summary of visual cortex excitability and visual imagery strength

In two separate experiments, we found that resting levels of early visual cortex excitability/activity negatively predicted visual imagery strength (fMRI and TMS, *Figures 2* and *3*). We were also able to causally alter visual imagery strength in two separate tDCS experiments. Specifically, decreasing

visual cortex excitability (using cathodal stimulation 1.5mA) increased imagery strength (see *Figure 4B and C*).

Our data suggest that visual cortex excitability plays a causal role in modulating imagery strength, but how exactly does excitability influence imagery strength? One hypothesis is that hyperexcitability might act as a source of noise in visual cortex that limits the availability or sensitivity of neuronal response to top-down imagery signals, resulting in weaker image-simulations. This hypothesis is supported by behavioral work showing that both imagery and visual working memory can be disrupted by the passive presence of uniform bottom-up afferent visual stimulation (*Keogh and Pearson, 2011*; *Keogh and Pearson, 2014*), known to increase neural depolarization in primary visual cortex (*Kinoshita and Komatsu, 2001*). However, the strength of the top-down imagery-signals arriving at visual cortex should also play a role in governing imagery strength, as activity in a brain network including prefrontal areas supports mental image generation (*Pearson et al., 2015b*). Therefore, we next assessed the role that prefrontal cortex activity plays in shaping visual imagery strength.

## Correlations between frontal cortex excitability and imagery strength

As mentioned previously, the exploratory, multiple comparison-corrected whole-brain surface-based analysis of the mean fMRI intensity levels at rest revealed relationships with clusters in both visual and frontal cortex (see *Figure 2* and *Figure 2—figure supplement 2* and Supplementary Table S2). Most of the significantly positive frontal clusters were located in superior frontal cortex. Additionally, using a ROI-based approach, normalized mean fMRI intensity levels in two frontal areas also showed positive relationships with imagery strength: superior frontal cortex ($r = 0.41$, $p=0.022$) and area parsopercularis ($r = 0.38$, $p=0.033$; ROIs defined by the Desikan–Killiany atlas). However, these relationships did not survive multiple comparison correction (both $p>0.05$).

## Manipulating prefrontal cortex excitability using tDCS

To explore the theoretical role of frontal cortex in imagery generation and maintenance further, we next sought to evaluate the effect of modulating neural excitability in prefrontal cortex using tDCS during image generation. The active electrode was placed between F3 and Fz (left frontal cortex), and the reference electrode on the right cheek (*Figure 5A* for montage). Participants completed both cathodal and anodal conditions (1.5mA) over 2 separate days. A linear mixed-effects analysis

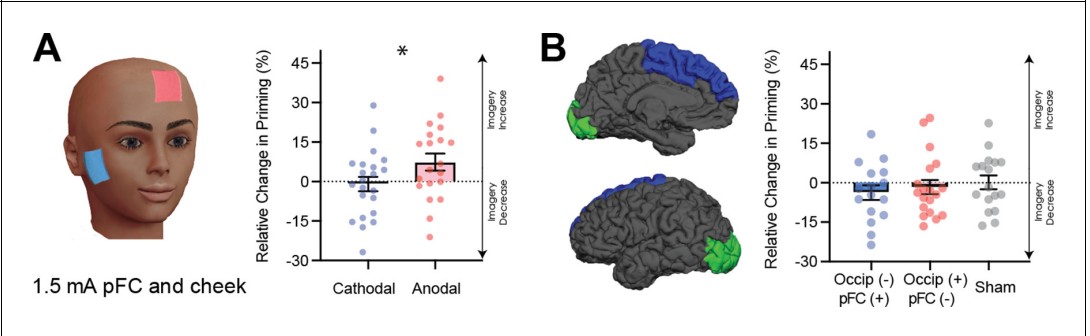

**Figure 5.** Data for prefrontal cortex stimulation. (**A**) Effect of left prefrontal (pFC) cortex stimulation on imagery strength at 1.5mA. The left image shows the tDCS montage, with the active electrode between Fz and F3 and the reference electrode on the right cheek. The right image shows the effect of cathodal (decrease excitability, blue dots represent each participant's difference score) and anodal (increase excitability, red dots represent each individual participant's difference score) stimulation averaged across all blocks during and after tDCS stimulation (D1, D2, P1, and P2). Imagery strength can be seen to increase with anodal stimulation. (**B**) Effect of joint electrical stimulation of prefrontal cortex and visual cortex. The left image shows brain areas targeted in the final tDCS study. Data shows non-significant effects of cathodal occipital + anodal pFC stimulation (blue bars, blue dots represent individual participants data), anodal occipital + cathodal pFC stimulation (red bars, red dots represents individual participants data) and sham stimulation (grey bars grey dots represent individual participants data). All error bars show ± SEMs and stars (*) indicate a significant effect of tDCS polarity.

The online version of this article includes the following source data for figure 5:

**Source data 1.** 1.5mA Prefrontal tDCS data.
**Source data 2.** 1.5mA combined tDCS data.

was run with a 2 (tDCS polarity: cathodal and anodal), x 4 (block: D1, D2, P1, P2 – see *Figure 4—figure supplement 1A* for timeline and *Figure 4—figure supplement 1G* for data for each block) x 2 (order of stimulation: cathodal on the first or second day) design. The effect of tDCS polarity was significant ($\chi^2(1)$=6.86, p=0.009, see *Figure 5A*). Interestingly, in contrast to the visual cortex, where decreasing excitability led to stronger imagery, we found the opposite pattern for frontal areas (see *Figure 5A*).

Taken together, these fMRI and tDCS data provide evidence that the cortical excitability of prefrontal cortex also plays a role in governing the sensory strength of visual imagery.

## The joint role of visual and frontal cortex activity in visual imagery strength: fMRI

Beyond the individual roles of prefrontal and visual cortex in forming mental images, evidence suggests that both areas can act together as part of an imagery network (*Østby et al., 2012*; *Schlegel et al., 2013*). Hence, we combined the whole-brain normalized mean fMRI intensity scores from the two areas (frontal and visual) and related their ratio to imagery strength. We found that the ratio of V1 to superior frontal activity predicted the strength of visual imagery (Spearman rank: $r_s = -0.53$, p=0.002). This effect also held when controlling for the Euklidean distance between the two areas (partial Spearman rank: $r_s = -0.54$, p=0.002). Hence, participants with both comparatively lower levels of visual cortex normalized mean intensity and higher frontal levels had stronger imagery.

To assess the possibility that cortical connectivity might be driving this fronto-occipital excitability relationship, we analyzed the individual functional connectivity of the same two areas for each participant, that is, the degree to which the BOLD signals in each area correlate over time. The functional connectivity did not significantly predict imagery strength (r = −0.24, p=0.19). This suggests that the combination of highly active frontal areas and low visual cortex excitability might present an optimal precondition for strong imagery creation, irrespective of the temporal coupling of their activity.

## The joint role of visual and frontal cortex activity in visual imagery strength: tDCS

To further investigate the possibility that optimal imagery strength occurs with a combination of low levels of excitability in visual cortex and high levels in prefrontal cortex, a new tDCS experiment was run where both prefrontal and visual cortex were simultaneously stimulated during imagery generation using the same blocked design as in all previous tDCS experiments (1.5mA). There were 3 conditions in this study, the first condition aimed to increase prefrontal (anodal) and decrease visual cortex (cathodal) excitability (blue dots in *Figure 5B*), the second condition decreased prefrontal (cathodal) and increased visual cortex (anodal) excitability (red dots in *Figure 5B*), and the third condition was a sham condition where the tDCS machine shut off after 30 s (grey dots in *Figure 5B*). A linear mixed-effects analysis was run with a 3 (tDCS polarity: cathodal, anodal and sham), x 4 (block: D1, D2, P1, P2 – see *Figure 4—figure supplement 2A* for timeline and *Figure 4—figure supplement 2E* for data for each block) x 3 (order of stimulation: cathodal on first, second or third day) design. However, the effect of tDCS polarity was not significant (tDCS $\chi^2(2)$=2.70, p=0.26), see *Figure 5B*).

## Summary of cortical excitabilities effect on imagery strength

In summary, visual cortex excitability reliably correlated negatively with the strength of visual imagery using both fMRI and TMS as measurement tools (*Figures 2* and *3*). Modulating visual cortex excitability also altered the strength of visual imagery (*Figure 4B & C*, see *Table 1* for a summary of all tDCS experiments). Specifically decreasing visual cortex excitability led to increased visual imagery strength. There was also evidence that altering prefrontal cortex excitability modulates visual imagery strength, but in the opposite pattern to visual cortex – increasing prefrontal cortex excitability led to increased imagery strength (see *Figure 5A*). However, combining stimulation of the frontal and occipital cortex had no reliable effect on modulating visual imagery strength.

**Table 1.** Summary of montage, intensity, duration, and significance of each tDCS experiment.

| Experiment # | Montage (EEG Coordinates) | Intensity + duration | Notes | Significant |
|---|---|---|---|---|
| 1 Occipital | Active: Inion (Oz) Reference: Supraorbital (Fpz) | 1 mA 15 min | Tested effect on imagery strength | × |
| 2 Occipital | Active: Inion (Oz) Reference: Right Cheek | 1.5 mA 15 min | Tested effect on imagery strength | ✓ |
| 3 Occipital | Active: Inion (Oz) Reference: Right Cheek | 1.5 mA 15 min | Tested effect on imagery strength (additional sham condition) | ✓ |
| 4 Prefrontal | Active: Between F3-Fz Reference: Right Cheek | 1.5 mA 15 min | Tested effect on imagery strength | ✓ |
| 5 Occipital + Prefrontal | Active: Inion (Oz) Active: Between F3-Fz | 1.5 mA 15 min | Tested effect on imagery strength | × |
| Additional control Occipital | Active: Inion (Oz) Reference: Right Cheek | 1.5 mA 15 min | Tested effect on phosphene threshold | ✓ |

## Discussion

Perhaps as far back as Plato, but overtly since the 1880s philosophers, scientists and the general populace have wondered why the human imagination differs so profoundly from one individual to the next. This question has recently gained fresh notability and attention with the introduction and classification of the term aphantasia to describe individuals who self-report no visual imagery at all (*Zeman et al., 2015*; *Keogh and Pearson, 2018*). Here, we provide evidence that pre-existing levels of neural excitability and spontaneous resting activity in visual cortex can influence the strength of mental representations as measured using the binocular rivalry paradigm. Our data indicate that participants with lower excitability in visual cortex have stronger sensory imagery. Furthermore, we provide causative evidence, using tDCS over visual cortex, that altering neural excitability in these areas can modulate imagery strength. Prefrontal cortex excitability also played a role in controlling the strength of visual imagery, however, in the opposite direction to visual cortex excitability.

It should be noted that while previous work has demonstrated that this measure of imagery strength can be separated from feature-based attention (*Pearson et al., 2008a*), we cannot explicitly rule out the possibility that tDCS was improving other cognitive mechanisms that are involved in this task such as sustained or selective attention. Visual imagery has also been shown to have multiple features such as strength, vividness, capacity and precision (*Keogh and Pearson, 2017*; *Pearson et al., 2011*; *Bergmann et al., 2016a*) . Here, we found that the strength of imagery was improved by cathodal stimulation of the early visual cortex and anodal of the prefrontal cortex. However, there was no evidence of visual or prefrontal cortex stimulation in altering the vividness of visual imagery, despite imagery strength correlating with these 'online' vividness measures. Additionally, our current studies also only used imagery of red and green Gabor patches, which have features particularly suited to early visual cortex and likely recruit this region during imagery. More complex imagery, such as imagery of faces, may rely less on representations in early visual cortex and more so on representations and excitability further upstream of the cortex, such as the fusiform face area in the case of face imagery. Future research should assess what qualities of visual imagery are, and are not, altered by stimulation of early visual cortex, and prefrontal cortex to further elucidate the neural mechanisms underlying individual differences in visual imagery. It may be the case that all forms of visual imagery are improved by prefrontal cortex stimulation, due to an increase in the strength of top-down signals, whereas only images that require activation of low-level sensory features, such as color or orientation, will be influenced by the excitability of early visual cortex.

Another limitation to our study is that although the majority of participants in our tDCS experiments showed the same pattern of results (larger increases in imagery strength in the cathodal vs anodal condition for occipital stimulation, and vice versa for prefrontal), there were some participants who showed the opposite pattern. It is important to note that while tDCS has been shown to modulate visual cortex excitability in numerous studies as well as our control experiment, there are large interindividual differences in the amount of the modulation that occurs for each individual (*Chew et al., 2015*; *López-Alonso et al., 2014*) and sometimes the direction of these excitability changes (*Strube et al., 2016*). Recent research suggests that cortical morphology (*Filmer et al.,*

2019a; *Laakso et al., 2019*; *Laakso et al., 2015*), proportion of neuro-modulators (*Filmer et al., 2019b*) and brain state (*Bergmann, 2018*) can all influence how well tDCS is able to modulate brain activity and behaviour.

Over the last 30 years, empirical work has demonstrated many commonalities between imagery and visual perception (see *Pearson et al., 2015b*; *Dijkstra et al., 2019* for a review). However, the two experiences have clear phenomenological differences between them. Our findings suggest a possible dissociation between mental imagery and visual perception in regards to cortical excitability's role in shaping externally versus internally driven visual representations. Previous work has demonstrated that perceptual sensitivity is associated with higher levels of visual cortex excitability (*Ding et al., 2016*; *Antal et al., 2001*; *Kraft et al., 2010*; *Reinhart et al., 2016*), whereas our results suggest the opposite for mental imagery; stronger imagery is associated with lower visual excitability. Interestingly, some studies of visual perception have found that reducing visual cortex excitability can improve performance on more complex perceptual tasks, such as discrimination and object tracking (*Antal et al., 2004a*; *Waterston and Pack, 2010*). It may be the case that, although both perception and imagery recruit visual cortex, the optimal visual cortex conditions for task performance vary as a function of task demands. Considering these results in terms of differences in signal-to-noise ratios (*Miniussi et al., 2013*) may help to explain the contrasting findings in perceptual tasks, as well as the results from our current imagery experiments.

Neural activity can be thought of as a combination of signal related activity and neural noise. The signal is often defined as the number, or proportion of neurons that code for a specific stimulus, with higher firing rates resulting in a stronger signal. Noise can be thought of as the activity of all other non-signal related neural activity. A higher signal-to-noise ratio will generally result in better performance on behavioral tasks. Increasing cortical excitability can potentially be a source of noise, through increasing the likelihood of all neurons in the stimulation region to fire, whereas decreasing cortical excitability reduces the likelihood of neurons to fire, reducing neural noise.

In very basic detection tasks, an injection of noise into the visual cortex may result in better detection by pushing a subthreshold signal over a given threshold, leading to a higher proportion of signal relative to noise. Conversely, in a discrimination task where two or more potential outcomes exist, it may be the case that adding random noise to the signal will enhance both (anticipatory) stimulus-related representations in a non-selective manner. Decreasing cortical excitability, on the other hand, reduces the likelihood of neurons to fire, which will decrease both anticipatory signals and neural noise, potentially resulting in better performance as only one of the two representations reach supra-threshold.

Decreasing visual cortex excitability during imagery may make it harder to induce an action potential in visual cortex neurons, through lowering membrane potentials. However, this also reduces random noise, which may result in a better signal-to-noise ratio and as such stronger imagery. Another possibility is that when tasks rely more heavily on top-down signals, such as in an imagery task, reduction in sensory noise might allow for better communication among neurons in the visual cortex. Furthermore, it is also possible that tDCS stimulation has a tendency to selectively reduce non-imagery related signals: tDCS appears to have a larger impact on the more superficial cortical layers than on the deeper cortical layers, as the superficial layers are closer to the current source (*Komarov et al., 2019*). Interestingly, recent research suggests that imagery-related signals are predominantly found in deep cortical layers (*Bergmann et al., 2019*). As a consequence, tDCS may attenuate signals in the mid- and superficial layers more than those in the deep layers, thereby causing a relative advantage of deep-layer imagery-related signals over the ones arising in the other layers.

The findings that visual cortex excitability is negatively related to imagery strength could hence be explained by hyperexcitability acting as a source of noise, which, when reduced, leads to less neural noise in the visual cortex resulting in a higher signal-to-noise ratio and thus stronger imagery. A signal-to-noise ratio explanation also aligns well with our findings related to prefrontal cortex excitability. Greater excitability of frontal cortex may allow for amplification of the top-down signal, either through boosting firing, or shaping of neuronal population activity. Increased top-down signals might also allow for a greater inhibition of non-signal related neural noise further down the cortical hierarchy, resulting in a higher signal-to-noise ratio in the visual cortex.

This signal-to-noise hypothesis of visual imagery is in line with findings from related research. A study on grapheme-color synesthesia found that - contradictory to our results - synesthetes had

enhanced resting-state visual cortex excitability (measured using phosphene thresholds). However, they also found that synesthetic experience could be enhanced by reducing visual excitability via tDCS (*Terhune et al., 2015a*). These seemingly contradictory results were thought to be due to two different mechanisms. The authors suggested that a hyperexcitable visual cortex during brain development may be what leads to individuals developing synesthesia in the first place; in adulthood, however, decreasing visual cortex excitability might lead to increased signal-to-noise in the visual cortex, thereby enhancing the synesthetic experience (*Terhune et al., 2015a*). In addition to this, other research indicates that the expectation of a visual stimulus leads to a stimulus template in visual cortex, with reduced activity in V1 and improved stimulus decoding by pattern classifiers (*Kok et al., 2013*). Similarly, reduced early visual cortex activity increases the likelihood of visual hallucinations in a subsequent detection task (*Pajani et al., 2015*). The convergence of these data appears to indicate that 'background' neural noise in sensory cortices may play an important role in modulating the strength of mental representations.

Despite much evidence for the involvement of the prefrontal cortex and visual cortex working in concert during visual imagery, we found that while manipulating either prefrontal or visual cortex excitability in isolation could induce increases in imagery strength, simultaneous stimulation of visual and prefrontal cortices had no effect on visual imagery. One possible explanation for these results is that modulating activity in two regions of the brain is too much of a change and has an overall disruptive effect on imagery formation. However, neither of the stimulation conditions resulted in significant reductions as compared to sham stimulation, making this explanation unlikely.

There also exists large variability in prefrontal cortex anatomy and tDCS effectiveness, with recent research showing that the thickness of left prefrontal cortex correlated with behavioral changes from anodal (but not cathodal) stimulation (*Filmer et al., 2019a*). It might be that these large variations play a role in our null findings; however, we did find that isolated stimulation to prefrontal cortex modulated imagery strength, making this another unlikely explanation of these null results. A plausible explanation may be that during simultaneous stimulation of visual and prefrontal cortex, other regions were modulated as well, inducing the null effect (*Bikson et al., 2010*), or that this montage leads to smaller current densities and changes in excitability in both visual and prefrontal cortex. For example, a previous study found evidence that the distance between electrodes alters the stimulation effects when other stimulation parameters are kept consistent (*Moliadze et al., 2010*). It might be the case that due to the spacing of the electrodes the current density may have been reduced, as the reference electrode was further from the active electrode as compared to our studies with significant results (Supraorbital vs cheek in significant studies). Our first study also resulted in non-significant results, which may be due to a lower intensity of stimulation (1mA vs 1.5mA in significant studies). However, this may also have been driven in part by the placement of the reference electrode. It seems possible that in our case, the montage we used to stimulate prefrontal and occipital cortex simultaneously may not have been sufficient to alter cortical excitability in these two regions, resulting in no significant changes in visual imagery strength.

Our findings do conflict with some previous research on visual imagery. For example, one study found that applying 1 Hz TMS to area BA 17 (primary visual cortex), slowed responses in a task where individuals had to imagine stripes (or were perceptually shown stripes) and answer questions about these images (*Kosslyn, 1999*). Although these chronometry type experiments are very common in early visual imagery research and were important in advancing the field as a whole, they do not provide any information about the quality or sensory representational nature of the images held in mind. Slower reaction times on both the perception and imagery task may be due to a general slowing of cognitive performance or visual scanning, rather than reflecting any change in the quality of the visual images created in the mind. Previous work has also found positive correlations between BOLD activity in the visual cortex and the vividness of visual imagery questionnaire (*Cui et al., 2007*; *Amedi et al., 2005*). Additionally, some TMS studies have found that during visual imagery, visual cortex excitability increases (*Cattaneo et al., 2011*; *Sparing et al., 2002*). These findings at first may seem incompatible with our results; however, these studies measure event-related neural changes, rather than comparing changes in task performance due to modulation of neural activity, or assessing how the resting levels of visual cortex activity influence task performance. It may very well be the case that on average in our tasks neural activity increases with imagery, and perhaps those with the lowest levels of resting activity have the largest changes in neural activity. For example, to calculate BOLD changes a baseline of 'resting-state' activity is used. It may be that participants with initially

low visual cortex excitability are able to increase visual cortex activity more-so than those with higher levels, and this could potentially explain the larger BOLD changes for individuals with stronger visual imagery.

It is possible that the observed effects of cortical excitability may be driven by individual differences in inhibitory and excitatory neurotransmitter concentrations. Numerous studies have investigated what neurotransmitters modulate cortical excitability with GABA and Glutamate being implicated in controlling inhibition and excitability, respectively. While the relationship between GABA and cortical excitability/activity is more ambiguous (*Terhune et al., 2015b*; *Boillat et al., 2020*), the concentration of glutamate in the early visual cortex has been shown to correlate positively with visual cortex excitability (measured by phosphene thresholds) in both normal and synesthetic participants (*Terhune et al., 2015b*). There is also evidence for a strong link between BOLD-fMRI activity and glutamate concentration: using a combined fMRI-MRS approach where BOLD-fMRI activity and glutamate signals were recorded simultaneously, researchers found that the time courses of fMRI-BOLD activity and Glutamate concentration were strongly correlated (*Ip et al., 2017*). Evidence of such a relationship at a between-subject level is missing but seems plausible. If this is the case, then the observed relationships of our neural measures and visual imagery may (at least partly) be due to individual differences in the concentration of glutamate in visual cortex: a lower level of glutamate in the visual cortex might result in less excitatory neuronal noise, thereby increasing the signal-to-noise ratio of top-down signals that govern the generation of internal images in the visual cortex.

A plethora of imagery research has demonstrated evoked and content specific BOLD responses in early and later visual cortex when individuals form a mental image (for reviews of this work see: *Pearson et al., 2015b*; *Dijkstra et al., 2019*). Here, however, we took a different approach by examining the individual variation in brain physiology that might form the preconditions for strong or weak imagery. This endeavor required a non-event related design. Interestingly, such non-event related designs utilizing inter-individual differences are now commonly used to mechanistically link human cognition and brain function or anatomy (*Kanai and Rees, 2011*). Our results add to this growing body of research, which demonstrates that pre-existing brain activity parameters can fundamentally influence mental performance.

Our observations may also have clinical applications: In many mental disorders, imagery can become uncontrollable and traumatic. On the other hand, mental imagery can also be harnessed specifically to treat these disorders (*Pearson et al., 2015b*). Interestingly, disorders that involve visual hallucinations such as schizophrenia and Parkinson's disease are both associated with stronger and/or more vivid mental imagery (*Shine et al., 2015*; *Sack et al., 2005*). It has recently been suggested that the balance between top-down and bottom-up information processing may be a crucial factor in the development of psychosis, with psychosis prone individuals displaying a shift in information processing towards top-down influences over bottom-up sensory input (*Teufel et al., 2015*). Our data indicate that it may be possible to treat symptomatic visual mental content by reducing its strength via non-intrusively manipulating cortical excitability. Alternatively, we may be able to 'surgically' boost mental image simulations specifically during imagery-based treatments, resulting in better treatment outcomes. Further research on longer lasting stimulation protocols, and the individual differences in response to brain stimulation is needed to assess its therapeutic potential.

In conclusion our data demonstrates that visual cortical excitability, as well as prefrontal excitability, appears to play a role in governing the strength of an individual's visual imagery strength providing a potential explanation for the large variation in visual imagery that exists within the general population and providing a promising new tool for altering the strength of visual imagery.

## Materials and methods

### Study design

The first study with fMRI was exploratory, to assess whether resting levels of BOLD might predict visual imagery. We followed this with a correlational TMS study and aimed to collect phosphene thresholds from 30 to 35 participants, which would give us power of around 80–85% for a moderate correlation (in line with the fMRI correlations of r = ~. 45). All tDCS experiments were designed as repeated measures studies (with the aim of all participants completing all conditions in the study). We aimed to collect data from 15 to 20 participants, as most tDCS studies examining effects on

cognition have found significant effects with this range of participants (see for examples: *Javadi and Cheng, 2013*; *Strobach et al., 2015*; *Manuel et al., 2014*; *Javadi et al., 2012*). Data collection stopped once we had at least 15 participants in each group who had completed 2 days of testing, no more participants were recruited beyond this point; however, participants were not cancelled if we reached 15 (e.g. if we had collected 15 participants for both days but still had 2 more participants who had completed 1 day of testing, we still ran them through the study – resulting in 17 participants).

## Participants

All MRI participants were right-handed and had normal or corrected-to-normal vision, with no history of psychiatric or neurological disorders. All tDCS and TMS participants had normal or corrected-to-normal vision, with no history of psychiatric or neurological disorders, as well as no history of migraines and/or severe or frequent headaches. All MRI research was carried out in Germany at the Max Planck Institute for Brain Research and all brain stimulation research (tDCS and TMS) was carried out in Australia at the University of New South Wales. Written informed consent was obtained from all participants and the ethics committee of the Max Planck Society approved the MRI study and the ethics committee of the University of New South Wales approved the tDCS and TMS studies.

## Exclusion criteria for the tDCS experiments

There were a number of strict exclusion criteria chosen a priori for the tDCS experiments due to the technical psychophysics and brain stimulation experiments involved here. These are based on previous work using the binocular rivalry paradigm, which is a sensitive measure of visual imagery strength when participants complete the task correctly. Due to the nature of the task, it is important to include catch trials/exclusion criteria to assess whether participants are correctly and reliably completing the task. These values are based on exclusion criteria we have used in previous experiments using this paradigm.

## Brain imaging sample

Thirty-two individuals (age range: 18–36 years, median: 25.5; 13 males) participated in the fMRI resting-state and retinotopic mapping measurements and in the behavioral experiment. These individuals had been part of previous studies (*Bergmann et al., 2016a*; *Bergmann et al., 2016b*). Of the original imagery study (*Bergmann et al., 2016a*), which included 34 participants, 1 participant had not done the fMRI resting-state measurement (but this individual participated in the additional fMRI resting-state measurement, see further below). The other participant was excluded because of reporting several migraine attacks shortly prior to the measurement. Migraine is known to affect fMRI BOLD activity and cortical excitability (*Coppola and Schoenen, 2012*; *Welch, 2005*). One participant was excluded from data analysis because of misunderstanding the task instructions in the behavioral imagery task (this participant had already been excluded in the original study). The data analysis was done with the remaining 31 individuals. To look at the reliability of the observed relationships, we also analysed the data of an additional fMRI resting-state measurement (with different sequence parameters, see further below). This sample also included 31 individuals (age range: 22–36 years). Of these, 30 had also participated in the original resting-state measurement. Hence, there were two participants of which only one resting-state measurement was available, respectively: for one participant, only the fMRI measurement with TR = 2 s had been done, and for the other one, only the fMRI measurement with TR = 1 s. Participants were reimbursed for their time at a rate of 15€ per hour. Written informed consent was obtained from all participants and the ethics committee of the Max Planck Society approved the study.

## TMS samples

All participants in both the TMS and tDCS studies had normal or corrected to normal vision, no history of any neurological or mental health issues or disorders, no history of epilepsy or seizures themselves or their immediate family, no history of migraines and no metal implants in the head or neck region. We aimed to collect phosphene thresholds from 30 to 35 participants, which would give us power of around 80–85% for a moderate correlation (in line with the MRI correlations). A total of 37

participants participated in this study for money ($30 per hour) or course credit, five participants were excluded due to an inability to produce reliable phosphenes (see *Table 2* and exclusion criteria). Of the remaining thirty-two participants 15 were female, age range: 18–30). Written informed consent was obtained from all participants and the ethics committee of the University of New South Wales approved the study.

## TDCS samples

For all tDCS experiments, we aimed to collect data from 15 to 20 participants, as most tDCS studies examining effects on cognition have found significant effects with this range of participants (see for examples: *Javadi and Cheng, 2013*; *Strobach et al., 2015*; *Manuel et al., 2014*; *Javadi et al., 2012*). A priori we chose a cut-off of 33% of trials being mixed as an exclusion criterion (see *Table 2* for explanations of all exclusion criteria and *Table 3* for tally of all exlcuded participants). The reason for this is that mixed trials are not included in our analysis, and as such a large number of mixed trials vastly decrease the number of analyzable trials. Participants whose imagery scores were lower than 40% were also excluded, as it is difficult to tell if these data should be defined as strong or weak imagery, or due to a participant just not completing the task correctly. This measure of imagery strength is predicated on how the energy of a stimulus impacts on binocular rivalry. Very weak perceptual stimuli prime binocular rivalry up unto a certain point. At this tipping point, as the image becomes stronger, it begins to suppress binocular rivalry (*Brascamp et al., 2007*). For this reason, when priming is low this indicates suppression that may either mean the participant's visual imagery is so strong it is suppressing binocular rivalry, or the participant is not completing the task correctly.

For the first tDCS experiment (1mA, Occipital and Supraorbital), a total of 21 subjects participated for money or course credit. Five participants were excluded from our analysis, as the number of usable trials was small due to too many reported mixed rivalry percepts (more than a third of trials, N = 3) and two were excluded due to low priming scores (see *Table 1* for exclusion criteria). Of the remaining 16 participants, 7 were female, and the age range was 18–32.

For the second tDCS experiment (1.5mA, Occipital and Cheek), a total of 37 subjects participated for course credit. Due to a faulty tDCS cable, many of these participants had the tDCS machine shut off/exceeded voltage on the first day of testing (N = 9, see *Table 2* for exclusion criteria) and they did not come back for the second day of testing. Of the 30 participants who did not exceed voltage in the first day two were excluded due to low priming and two had a high number of mixed trials (see *Table 1* for exclusion criteria). Of the remaining 26, nine participants only had one day of testing data available due to the machine exceeding voltage on the second day of testing, two participants had one day of testing removed due to low priming on one of the 2 days, age range 18–26.

For the third tDCS experiment (1.5mA, Occipital and Cheek + Sham), a total of 28 subjects participated for course credit or payment ($40 per hour). Two participants were excluded due to technical issues with the computer on the first day of testing, one participant was excluded for pressing the incorrect buttons during the task/due to misunderstanding of the task, one participant was excluded due to removing the tDCS during the testing session, one participant was removed for a high number of mixed trials and one for high mock priming (see *Table 1* for exclusion criteria). Of the remaining 22 participants, the age range was 18–23. Of these participants 4 only completed 2 of the 3 days of testing due to attrition (N = 3) and machine malfunction/exceeded voltage (N = 1). Another 4 participants also only completed one day of testing due to attrition (N = 3) and machine malfunction (N = 1).

For the fourth tDCS experiment (1.5mA, left prefrontal and cheek), 31 participants participated in the study for course credit. Due to a faulty tDCS cable, for many of these participants the tDCS machine shut off/exceeded voltage on the first day of testing (N = 4), and as such, they did not complete the study. Two participants' data was removed due to very high mock priming, indicating either a misunderstanding of the task or demand characteristics and one was removed due to very low priming (see *Table 1* for exclusion criteria). Of the remaining 24 participants, seven had one day's worth of data due to the tDCS machine shutting off/exceeding voltage (N = 3) or attrition (N = 2), or having very low imagery priming on one of the days (N = 2), age range was 18–25 years.

For the fifth tDCS experiment (1.5mA, left prefrontal and occipital cortex + sham), 28 participants participated in the study for course credit or payment (AUD $40 per hour). Three participants were excluded due to very low priming, one participant was excluded due to technical issues with the tDCS machine (exceeding impedance/voltage on the first day) and five participants were excluded

**Table 2.** Exclusion criteria for tDCS experiments.

| Exclusion | Explanation |
| --- | --- |
| Mock Priming (Higher than 66%) | Mock displays are fake binocular rivalry displays – priming on these trials indicates that participants are showing a response/demand characteristic and as such we cannot trust their priming scores, as they may either be responding in a way that they think we want them too, or they are not attending to the task correctly. A score of more than 66% indicates that the participant has primed on these mock trials more than once. |
| Low Priming (lower than 40%) | Participants whose imagery scores were lower than 40% were excluded, as the score becomes difficult to interpret: The measure of imagery strength is predicated on how the energy of a stimulus impacts on binocular rivalry. Very weak perceptual stimuli prime binocular rivalry up unto a certain point. At this tipping point, as the image becomes stronger, it begins to suppress binocular rivalry (*Brascamp et al., 2007*). For this reason, very low priming can either mean that the participant's visual imagery is so strong that it leads to suppression, or that the opposite is the case, and imagery is in fact very weak. Alternatively, it may also be that a participant is not completing the task correctly. 40% was chosen as the cut off as this is 10% lower than chance values (50%). |
| Mixed Percepts (higher than 1/3 or 33%) | We analyse our priming data as percent primed, that is the percentage of trials a participant imagined an image, then saw that image in the following display. Mixed trials cannot be labelled as either 'primed' or 'not primed', and as such these trials are not included in the analysis. A high number of mixed trials reduces the number of usable trials for analysis. This can lead to large changes that may be spurious (much larger percentage changes due to a single trial) and not due to the stimulation parameters. We excluded an individual's data set if more than a third of the trials were mixed percept's. |
| Attrition | Attrition occurred when a participant did not turn up to or cancelled the second and/or third day of testing. |
| Impedance (Exceeding voltage: impedance greater than 55 kΩ) | For safety reasons, our tDCS machine was programmed to shut off once the impedance exceeded 55 kΩ (this is pre-programmed into the tDCS machine). As the participants completed the experiment in a blackened room by themselves we could not monitor the impedance of the machine in real-time and as such the machine could switch off during a block of the experiment. As we cannot know how much stimulation this participant received we are unable to use them reliably in our analysis – as different stimulation durations will lead to different excitability changes. The second (1.5mA Occip + cheek) and fourth (1.5mA pFC + cheek) tDCS experiments were run at the same time. There were a large number of cases of impedance being exceeded in these studies – it was discovered that this was due to a faulty wire, which was replaced halfway through the experimental data collection. |
| Phosphene Determination | If participants reported phosphenes in the wrong visual hemifield (e.g. left visual hemisphere was stimulated and participants reported phosphenes in the left visual hemifield) their data was excluded. Participants who had very large eye-blinks in the REPT procedure were excluded from the experiment, due to this potentially resulting in a missed phosphene leading to incorrect phosphene threshold estimation. |

**Table 3.** Number of participants excluded per exclusion criteria for tDCS experiments.

| Exclusion criteria | Exp 1 | Exp 2 | Exp 3 | Exp 4 | Exp 5 | Exp 6 | Total |
|---|---|---|---|---|---|---|---|
| Mock priming | | | 1 | 2 | 2 | | 5 |
| Low priming | 2 | 2 | | 1 | 3 | | 8 |
| Mixed percept's | 3 | 2 | 1 | | | | 6 |
| Attrition | | | | | | 1 | 1 |
| Impedance | | 9 | | 4 | 1 | 2 | 16 |
| Incorrect buttons | | | 1 | | 3 | | 4 |
| Technical issues | | | 3 | | | 2 | 5 |
| Phosphenes | | | | | | 6 | 6 |
| Total | 5 | 13 | 6 | 7 | 9 | 11 | 51 |

due to misunderstandings or incorrectly completing the task (either pressing incorrect buttons (N = 3) or 100% priming for mock rivalry (N = 2), see *Table 1* for exclusion criteria). Of the remaining 19 participants, the age range was 18–35 years and of these participants, two participants one sham condition was not analsyed (one due to machine malfunction – exceeded impedance/voltage and one, one participant's sham condition was removed due to low priming), and three participants did not complete the occipital cathodal + prefrontal anodal condition (machine malfunction– exceeded impedance/voltage).

All subjects participated in these studies for course credit or money - $30 AUD per hour.

## tDCS modulation of phosphene thresholds control study

A total of 29 subjects participated in this study for money ($30 AUD per hour) or course credit. Of these 29 participants, 11 were excluded due to a number of strict exclusion criteria in regard to reliability of phosphene thresholds: If participants reported phosphenes in the wrong visual hemifield (e.g. left visual hemisphere was stimulated and participants reported phosphenes in the left visual hemifield) their data was excluded (N = 2), see *Table 1* for exclusion criteria. If participants blinked during the rapid estimation of phosphene thresholds (REPT) procedure their data was also removed from analysis (N = 4). A participant's data was also removed if the REPT procedure took longer than 5 min to set up after tDCS stimulation (N = 1). 3 participants were also removed due to technical issues with the tDCS machine exceeding voltages on one of the days (N = 2) or the REPT Matlab procedure experiencing errors (N = 1). One participant was removed due to attrition (N = 1). This resulted in 18 participants' data being analyzed (8 female, age range 18–25).

## TMS Phosphene threshold reliability study

This sample consisted of the same 29 subjects that participated in the control study to test tDCS modulation of phosphene thresholds. Exclusion criteria were the same as stated above, with the exception that those participants who had technical issues with the tDCS machine were included as their pre-tDCS TMS phosphene values were still usable (N = 2); the study also included 1 participant who took longer than 5 min to set up the TMS after the tDCS stimulation. This resulted in 21 participants' data being included in this correlation.

## Behavioral measurements

### Apparatus

### Brain imaging sample

Participants sat in a darkened room with dark walls, wearing red-green anaglyph glasses for the binocular rivalry imagery paradigm. Their head position was stabilized with a chin rest and the distance to the screen was 57 cm. The stimuli were presented on a CRT monitor (HP p1230; resolution, 1024 × 768 pixels, refresh rate: 150 Hz; visible screen size: 30° x 22.9°) and controlled by MATLAB R2010a (The MathWorks, Natick, MA) using the Psychophysics Toolbox extension (*Brainard, 1997*; *Pelli, 1997*; *Kleiner et al., 2007*), running on Mac OSX, version 10.7.4.

### TMS/tDCS sample

All experiments were performed in a blackened room on a 27 inch iMac with a resolution of 2560 × 1440 pixels, with a frame rate of 60 Hz. A chin rest was used to maintain a fixed viewing distance of 57 cm. Participants wore red-green anaglyph glasses throughout all experiments.

### Stimuli

### Brain imaging sample

The circular Gaussian-windowed Gabor stimuli were presented centrally, spanning a radius of 4.6° around the fixation point in visual angle (thereby covering a diameter of 9.2°), one period subtending a length of 1.2°. The peak luminance starting value was ~0.71 cd/m$^2$ for the red horizontal grating, and ~0.73 cd/m$^2$ for the green vertical grating, which was then individually adjusted for each participant to compensate for eye dominance (see further below).

### TMS/tDCS sample

The binocular rivalry stimuli were presented in a Gaussian-windowed annulus around the bull's eye and consisted of a red-horizontal (CIE X = 0.579 Y=0.369 and green-vertical (CIE X = 0.269 Y=0.640), Gabor patch, one cycle/°, with a diameter of 6° and a mean luminance of 6.06 cd/m$^2$. The background was black throughout the entire experiment.

### Mock trials

Mock rivalry displays were presented on 10% of trials in the behavioral measurements with the brain imaging sample, 25% of trials in the first two tDCS experiments as well as the TMS experiment, and in 12.5% of the last three tDCS experiment to assess demand characteristics. The mock displays consisted of a spatial mix of a red-horizontal and green-vertical Gabor patch (50/50% or 25/75%). The mock display was spatially split with a blurred edge and the exact path of the spatial border changed on each trial based on a random function. Otherwise, the mock rivalry displays had the same parameters as the Gabor patches described in the previous paragraph.

## Procedure

All participants first underwent a previously documented eye dominance task (*Pearson et al., 2008a*). They then underwent the binocular rivalry imagery paradigm which has been shown to reliably measure the sensory strength of mental imagery through its impact on subsequent binocular rivalry perception (*Pearson, 2014*; *Pearson et al., 2008a*; *Keogh and Pearson, 2011*; *Keogh and Pearson, 2014*; *Pearson and Brascamp, 2008b*; *Sherwood and Pearson, 2010*; *Rademaker and Pearson, 2012*; *Pearson et al., 2015b*), thus avoiding any reliance on self-report questionnaires or compound multi-feature tasks. Previous work has demonstrated that when individuals imagine a pattern or are shown a weak perceptual version of a pattern, they are more likely to see that pattern in a subsequent brief binocular rivalry display (see *Pearson, 2014* for a review). Longer periods of imagery generation or weak perceptual presentation increase the probability of perceptual priming of subsequent rivalry (*Pearson et al., 2008a*; *Pearson et al., 2015b*; *Brascamp et al., 2007*). For this reason, the degree of imagery priming has been taken as a measure of the sensory strength of mental imagery (*Pearson, 2014*; *Pearson et al., 2008a*; *Keogh and Pearson, 2011*; *Keogh and Pearson, 2014*; *Pearson and Brascamp, 2008b*; *Sherwood and Pearson, 2010*; *Rademaker and Pearson, 2012*; *Pearson et al., 2015b*). This measure of imagery strength has been shown to be both retinotopic location- and spatial orientation-specific (*Bergmann et al., 2016a*; *Pearson and Brascamp, 2008b*), is closely related to phenomenal vividness (*Rademaker and Pearson, 2012*; *Chang et al., 2013*), is reliable when assessed over days (*Rademaker and Pearson, 2012*) or weeks (*Bergmann et al., 2016a*), is contingent on the imagery generation period (therefore not due to any rivalry control *Sherwood and Pearson, 2010*) and can be dissociated from visual attention (*Pearson et al., 2008a*).

At the beginning of each trial of the imagery experiment, participants were presented with a letter 'R' or 'G' which indicated which image they were to imagine (R = red horizontal Gabor patch, G = green vertical Gabor patch). Participants then imagined the red or green pattern for either 6 (tDCS and TMS experiments) or 7 s (behavioral measurements of the brain imaging sample). Following this imagery period, the binocular rivalry display appeared for 750 ms and participants indicated

which image was dominant by pressing '1' for mostly green, '2' for a mix and '3' for mostly red. During the behavioral measurements of the brain imaging sample and in both the 1.5 ma occipital tDCS experiments, as well as the 1.5ma pFC tDCS experiment, on-line ratings of imagery vividness were collected by having participants rate the vividness of the image they had created (on a scale of '1'=least vivid to '4'=most vivid) on each trial after the imagery period and before the binocular rivalry display.

For the tDCS experiments, there were no effects for mean subjective ratings of imagery vividness (see Supplementary *Figure 1—figure supplement 1C*). For the subjective vividness ratings acquired in the brain imaging sample, we conducted a whole-brain surface-based analysis of the fMRI resting-state data (see Materials and methods and *Figure 1—figure supplement 1B* and Table S3). Throughout all imagery experiments, participants were asked to maintain fixation on a bulls-eye fixation point in the center of the screen.

## Brain imaging sample

Participants completed 100 trials of the standard imagery paradigm per session (outside the scanner). The behavioral test session was repeated after an average of ~2 weeks with each participant. All of the runs were divided into blocks of 33 trials, and participants were asked to take a rest in between. In one participant, there was a strong perceptual bias for one of the two rivalry patterns in the first session due to incorrectly conducted eye dominance adjustments. Therefore, only the data set from the second session of this participant was used for later analysis. The retests demonstrated a very high retest reliability of the imagery strength measure ($r$ = 0.877, p<0.001). The data from both conditions were checked for normal distribution using Shapiro-Wilk normality test. No violation of the normality assumption was detected (both p>0.52).

## TMS/tDCS samples

For the TMS study, participants completed one block of 40 imagery trials. In the first tDCS experiments comprising of cathodal and anodal conditions, participants completed a total of 40 trials for each block resulting in a total of 480 trials across the two days of testing. In the tDCS experiments with cathodal, anodal and sham conditions participants completed 48 trials per block, and as such 864 trials were completed.

## Control tDCS modulation of phosphene thresholds experiment

Participants completed both the anodal and cathodal stimulation across two days separated by at least 24 hr, the order of which was randomized and counterbalanced across participants. Participants completed a memory or psychophysical task (both of which are not relevant to the current study) followed by the automated REPT phosphene threshold procedure prior to tDCS stimulation. Following this, participants completed two blocks of the imagery task (see main tDCS methods for full description of procedure and stimuli) with 15 min of cathodal or anodal stimulation. Immediately after the tDCS stimulation participants completed the automated REPT procedure again.

## Neuroimaging experiments

All neuroimaging data were acquired at the Brain Imaging Center Frankfurt am Main, Germany. The scanner used was a Siemens 3-Tesla Trio (Siemens, Erlangen, Germany) with an 8-channel head coil and a maximum gradient strength of 40 mT/m. Imaging data were acquired in two or three scan sessions per participant.

## Anatomical imaging

For anatomical localization and coregistration of the functional data, T1-weighted anatomical images were acquired first using an MP-RAGE sequence with the following parameters: TR = 2250 ms, TE = 2.6 ms, flip angle: 9°, FoV: 256 mm, resolution = 1×1 x 1 mm$^3$.

## fMRI retinotopic mapping measurement and analysis

This procedure has already been described in previous studies (*Bergmann et al., 2016a*; *Bergmann et al., 2016b*; *Genç et al., 2015*); the retinotopic maps acquired in studies [20, 22] were used for the ROI-based fMRI resting-state analyses in the present study. A gradient-recalled echo-

planar (EPI) sequence with the following parameter settings was applied: 33 slices, TR = 2000 ms, TE = 30 ms, flip angle = 90°, FoV = 192 mm, slice thickness = 3 mm, gap thickness = 0.3 mm, resolution = $3 \times 3 \times 3$ mm$^3$. A MR-compatible goggle system with two organic light-emitting-diode displays was used for presentation of the stimuli (MR Vision 2000; Resonance Technology North-ridge, CA), which were generated with a custom-made program based on the Microsoft DirectX library (*Muckli et al., 2005*). The maximal visual field subtended 24° vertically and 30° horizontally.

### Retinotopic mapping procedure

To map early visual cortices V1, V2, and V3, our participants completed two runs, a polar angle mapping and an eccentricity mapping run. The rationale of this approach has already been described elsewhere (*Sereno et al., 1995*; *Wandell et al., 2007*). Polar angle mapping: For the mapping of boundaries between areas, participants were presented with a black and white checkerboard wedge (22.5° wide, extending 15° in the periphery) that slowly rotated clockwise around the fixation point in front of a grey background. In cycles of 64 s, it circled around the fixation point 12 times at a speed of 11.25 in polar angle/volume (2 s). Eccentricity mapping: To map bands of eccentricity on the corti-cal surface to the corresponding visual angles from the center of gaze, our participants were pre-sented with a slowly expanding flickering black and white checkerboard ring in front of a grey background (flicker rate: 4 Hz). The ring started with a radius of 1° and increased linearly up to a radius of 15°. The expansion cycle was repeated 7 times, each cycle lasting 64 s. The participants' task in both mapping experiments was to maintain central fixation.

### Retinotopic mapping data analysis

We used FreeSurfer's surface-based methods for cortical surface reconstruction from the T1-weighted image of each participant (*Dale et al., 1999*; *Fischl et al., 1999*) (http://surfer.nmr.mgh.harvard.edu/fswiki/RecommendedReconstruction). FSFAST was applied for slice time correction, motion correction and co-registration of the functional data to the T1-weighted anatomical image. Data from the polar angle and eccentricity mapping experiment were analysed by applying a Fourier transform to each voxel's fMRI time series to extract amplitude and phase at stimulation frequency. Color-encoded F-statistic maps were then computed, each color representing a response phase whose intensity is an F-ratio of the squared amplitude of the response at stimulus frequency divided by the averaged squared amplitudes at all other frequencies (with the exception of higher harmonics of the stimulus frequency and low frequency signals). The maps were then displayed on the cortical surface of the T1-weighted image. Boundaries of areas V1, V2 and V3 were then estimated manually for each participant on the phase-encoded retinotopic maps up to an eccentricity of 7.2°.

### fMRI Resting-state data acquisition and analysis

#### fMRI resting-state data acquisition

The fMRI resting-state data (TR2) were collected using a gradient-re- called echo-planar imaging (EPI) sequence with the following parameters: TR = 2000 ms, TE = 30 ms, flip angle = 90°, FoV = 192 mm, slice thickness = 3 mm, number of slices = 33, gap thickness = 0.3 mm, voxel size = $3 \times 3 \times 3$ mm$^3$, acquisition time = 9 min, 20 s (thus, 280 volumes were collected)The additional fMRI measure-ment (TR1) used a gradient-recalled echo-planar imaging (EPI) sequence with the following parame-ters: TR = 1000 ms, TE = 30 ms, flip angle = 60°, FoV = 210 mm, slice thickness = 5 mm, number of slices = 15, gap thickness = 1 mm, voxel size = $3.28 \times 3.28 \times 5$ mm$^3$, acquisition time = 7 min, 30 s (i.e. 450 volumes). During the scans, the screen remained grey and participants had no further instruction but to keep their eyes open and fixate a cross in the center of the grey screen. Of those individuals who participated in both fMRI measurements, half completed the two resting-state meas-urements in the same session, whereas the other half completed the measurements on two different days.

#### fMRI resting-state data analysis

whole-brain surface-based group analysis. For a first assessment of the relationship between behav-ior and the fMRI data, we ran whole-brain analyses with the mean fMRI intensity data using a sur-face-based group analysis in FreeSurfer. Preprocessing of the functional data was done using FSFAST, which included slice time correction, motion correction and co-registration to the T1-

weighted anatomical image. No smoothing was applied, and the first two volumes (TR2 data) or four volumes (TR1 data) of the fMRI measurement were discarded. In a first-level analysis, each individual's average signal intensity maps were computed (which included intensity normalization) and nonlinearly resampled to a common group surface space (fsaverage), which allows for comparisons at homologous points within the brain. Following this, all subjects' data were concatenated and a general linear model fit to explain the individual behavioral data by the individual mean fMRI intensity levels was computed vertex-wise using an uncorrected threshold of $p<0.05$. Correction for multiple comparisons was done using a pre-cached Monte Carlo Null-Z simulation with 10 000 iterations and a cluster-wise probability threshold of $p<0.05$. In addition, as we had also collected subjective vividness ratings in the brain imaging sample (see Procedure), we also ran the equivalent whole brain analysis for the vividness ratings with the TR2 data set, using each individual's mean vividness (see *Figure 1—figure supplement 1B* and Supplementary Table S3). As already described in our previous study (*Bergmann et al., 2016a*), the subjective vividness values of two individuals were extreme, leading to a violation of the normal distribution assumption. As normality is necessary for the general linear model fit (Shapiro-Wilk normality test: $W(31) = .885$, $p=0.003$), the vividness ratings of these two individuals were excluded in the whole brain analysis.

## fMRI resting-state data analysis

ROI-based approach. The fMRI resting-state data were first preprocessed individually for each participant using the preprocessing steps implemented in FSL's MELODIC Version 3.10 (http://fsl.fmrib. ox.ac.uk/fsl/fslwiki/MELODIC), which included motion and slice time correction, high-pass temporal filtering with a cut-off point at 200 s and linear registration to the individual's T1 anatomical image and to MNI 152 standard space. No spatial smoothing was applied. The first two of the 280 volumes of the TR2 measurement were discarded to allow for longitudinal magnetization stabilization (of the TR1 measurement, the first 4 volumes of the 450 volumes were discarded); further ROI-based analyses confirmed that the pattern of significant results in visual cortex did not change when more volumes (4, 6 and 8) were discarded (see *Figure 2—figure supplement 3*). To compute fMRI mean intensity of the early visual cortex in each individual's subject space, delineations of the areas were first converted from anatomical to functional space in each individual. To ensure that the conversion had not produced overlaps between areas V1-V3, the volumes were subsequently subtracted from each other. Time courses of V1-V3 were then determined to compute their mean intensity across time. To determine mean intensity for other brain areas, we relied on the gyral-based Desikan–Killiany Atlas (*Desikan et al., 2006*). To ensure that there was no overlap between posterior atlas-defined areas and the retinotopically mapped early visual cortex, which would result in the mean intensity of these areas being partly computed from the same voxels, the volumes of the retinotopically mapped areas were also subtracted from the adjacent atlas-defined areas; the fMRI mean intensity of the atlas-defined areas was then determined from the remainder of these. The estimates of fMRI mean intensity of the atlas- and retinotopically mapped areas (*Fox and Raichle, 2007*) were normalized by subtracting the whole brain's mean intensity from the area's mean intensity, divided by the standard deviation of the whole brain's mean intensity.

Like the behavioral data, the normalized mean intensity values were checked for normal distribution using Shapiro-Wilk normality test. None of the retinotopically mapped early visual cortices showed a violation of the normal distribution (all $p>0.20$). Of the 34 atlas-defined areas, the normalized mean fMRI intensities of 4 areas showed a violation of the normal distribution assumption ($p<0.05$; fusiform, inferiortemporal, parstriangularis and postcentral area). For this reason, the relationships with behavior were also computed using Spearman rank correlations. Like with Pearson product moment correlations, none of the intensities of these areas had a significant relationship with behaviour (all $p>0.20$). The ratio of V1 and superior frontal mean intensities showed a violation of the normal distribution assumption ($W(31) = .919$, $p=0.022$) due to one extreme value (subject S8). Therefore, Spearman rank correlation ($r_s$) was used to compute the relationship with behavior. To further examine the possibility that temporal coupling between V1 and superior frontal cortex might account for their inverse relationship with behavior, we also computed each individual's functional connectivity of these two regions by calculating the time-wise correlation of their resting-state signals in each individual. As the functional connectivity data did not violate the normal distribution assumption (Shapiro-Wilk normality test, $p=0.50$), Pearson product moment correlation was used to

examine the relationship with behavior. To correct for multiple comparisons in the brain-behaviour relationships, we first ran permutation tests where we randomly shuffled the subject labels of one of the paired measures and re-computed the pairwise correlations (1000 permutations). p-Values of the correlations were then determined via the percentile of the real correlation in the distribution of correlations from all permutations. Following this, the p-values were adjusted for multiple comparisons using the False Discovery Rate (FDR)(*Yekutieli and Benjamini, 1999*; *Benjamini and Hochberg, 1995*).

## Control analyses to examine the effect of individual head motion

To rule out the possibility that factors like individual differences in head motion contributed to intensity variability, we re-analysed the data in two ways: (1) Partialling out overall head movement: we first computed a measure of Framewise Displacement (FD), which is a frame-wise scalar quantity of the absolute head motion from one volume to the next using the six motion parameters (*Power et al., 2012*). To obtain one value of absolute movement parameter for each individual, we determined the absolute sum of the displacement values across the whole run. We then used this parameter as a control variable in a Spearman partial correlation analysis (since the normality assumption was violated for overall head motion, $W(31) = .790$, p<0.001). (2) Exclusion of volumes with strong head movements ('scrubbing'): volumes with strong head movements have previously been shown to influence functional resting-state connectivity patterns (*Power et al., 2012*). To investigate whether such movement also influenced individual mean resting-state intensity in our sample, we re-analyzed our data as using the method described by Power et al. To identify affected timepoints (i.e. volumes), we first computed the Framewise Displacement (FD), and DVARS, which is a frame-wise measure of how much the signal changes from one volume to the next. Using Powell et al.'s threshold of 0.5 for FD and 0.5% signal change for DVARS, we then excluded all volumes that had FD and DVARS values above threshold, plus one frame back and two forward. After this procedure, we re-computed mean intensity for each participant and correlated the values with behaviour.

## Measuring and manipulating cortical excitability with brain stimulation (TMS and tDCS)

Cortical excitability can be broadly defined as the ease with which a neuron, or a population of neurons, can produce an action potential. The excitability of the brain is controlled by interactions between inhibitory (GABA acting upon $GABA_A$ and $GABA_B$ receptors) and excitatory (glutamate acting upon NMDA and non-NMDA receptors) neurotransmitters and cellular receptors. To measure cortical excitability and its role in visual imagery strength TMS (phosphene thresholds) were used, and non-invasive brain stimulation (in the form of transcranial direct current stimulation, tDCS) was employed to manipulate cortical excitability.

## Phosphene thresholds

Phosphene thresholds measure how much magnetic energy is needed to elicit an action potential and is inversely related to the excitability of the cortex, that is a more excitable cortex will require less magnetic energy to produce a phosphene. Many studies have used phosphene thresholds as a reliable measure of visual cortex excitability, for examples see: (*Cowey and Walsh, 2000*; *Stewart et al., 2001*; *Bestmann et al., 2007*; *Romei et al., 2008*). Phosphene thresholds have been shown to be reduced in populations that have elevated/altered cortical excitability, such as epilepsy and migraines (*Brigo et al., 2013*; *Afra et al., 1998*; *Aurora et al., 1998*; *Aurora and Welch, 1998*; *Young et al., 2004*). Studies of migraineurs who take pharmaceuticals to reduce cortical excitability as a treatment have demonstrated increased phosphene thresholds (reduced excitability) while taking these drugs (*Artemenko et al., 2008*; *Mulleners et al., 2002*; *Palermo et al., 2009*). A recent MRS study has also demonstrated that glutamate levels in the early visual cortex correlate negatively with phosphene thresholds (*Terhune et al., 2015b*). Taken together these results provide evidence that phosphene thresholds are a good proxy for measuring cortical excitability, although whether they measure inhibition, excitation or a combination of the two is still not fully known.

Transcranial Direct Current Stimulation (tDCS): Previous research has shown that tDCS can modulate motor and phosphene thresholds in a polarity specific way, such that anodal stimulation

increases cortical excitability while cathodal stimulation decreases excitability (*Nitsche et al., 2008*). In addition, tDCS to the visual cortex has been shown to affect visual evoked potentials (VEP's) in a polarity-specific manner, with cathodal stimulation reducing the amplitude of the N70 component, whereas anodal stimulation increases the amplitude (*Antal et al., 2004b*). Work has shown that stimulating the motor cortex also results in neurotransmitter regulation when measured using MRS. Specifically, anodal stimulation has been shown to reduce GABA (*Kim et al., 2014*; *Bachtiar et al., 2015*) while cathodal stimulation reduced glutamate (*Stagg et al., 2009*).

The mechanisms of change in cortical excitability that occur with tDCS are thought to be different during and after stimulation (*Stagg and Nitsche, 2011*). Changes during tDCS stimulation are thought to be due solely to changes in membrane potential, which make it easier or harder for neurons to produce an action potential (depending on the polarity of the stimulation). Whereas post-tDCS stimulation changes are thought to be very similar to long-term potentiation (increasing excitability) and long-term depression (decrease excitability). These changes likely rely on the way anodal and cathodal stimulation differentially displace Magnesium ions on the NMDA ion channels resulting in either a rise (anodal) or decrease (cathodal) in post-synaptic $Ca2+$ concentrations, although the exact mechanisms that underlie tDCS are still not completely known.

These data suggest that tDCS can manipulate cortex excitability in a polarity-specific manner, potentially through modulating GABA and glutamate concentrations.

## Phosphene threshold determination

Phosphene thresholds were obtained using single pulse TMS with a butterfly shaped coil (Magstim $\text{Rapid}^2$, Carmarthenshire, UK). The coil was placed centrally and approximately 2 cm above the inion. To obtain each participant's phosphene threshold, we used the previously documented automated rapid estimation of phosphene thresholds (REPT) (*Abrahamyan et al., 2011*). This REPT procedure uses a Bayesian adaptive staircase approach to find the 60% phosphene threshold of each participant.

Before the REPT procedure, we first ensured that our participants were able to see reliable phosphenes. We initially placed the coil centrally and approximately 2 cm above the inion (with the coil oriented 45 degrees) and gave the participant's single pulses starting at 50% of the machines maximal output, moving up to 85% of the machines maximal output in 5% step increments. If the participant failed to report any phosphenes (or if the location of the coil was producing very large eye blinks or neck twitches) the coil was moved approximately 1 cm left and the same procedure occurred. If this did not elicit phosphenes the coil was moved to the right-hand side (approximately 2 cm above the inion and 1 cm to the right) and the same procedure occurred. If the participant still could not see any phosphenes they did not complete any more of the experiment. While this initial testing took place participants were seated in front of a computer screen with a piece of black cardboard with white numbered quadrants covering the monitor. The participants were instructed to relax and stare forward at a fixation dot in the middle of the black cardboard and to let the experimenter know if they saw any sort of visual disturbances on the cardboard and if they did see something to describe what they saw and in which quadrant they saw it. If participants reported a phosphene occurring in an incorrect location, for example the left visual field when stimulating the left visual cortex, then they were excluded from the study and did not complete the REPT procedure.

During the REPT procedure participants were seated in front of the same computer screen as the initial phosphene testing with a piece of black cardboard with white numbered quadrants covering the monitor. The coil was placed in the same location as where phosphenes were elicited in the initial phosphene testing described above using a clamp attached to the testing table. Participants received 30 pulses, of varying intensities, which were delivered automatically by the machine when the participant pressed the space key (self-paced). After each pulse participants were instructed to indicate if they had seen a phosphene by pressing the left ('no I did not see a phosphene') or right ('yes I did see a phosphene') shift keys. After the REPT procedure, the experimenter asked the participant to report which quadrants the participant had seen the phosphenes in to ensure they were the same as the initial testing.

Transcranial direct current stimulation tDCS was delivered by a battery-driven portable stimulator (Neuroconn, Ilmenau, Germany) using a pair of 6 × 3.5 cm rubber electrodes in two saline-soaked sponges.

Four different montages were used across the different experiments. In experiment 1, the active electrode was placed over Oz while the reference electrode was placed over the midline supraorbital area (see *Figure 4A*). In experiments 2 and 3 and the phosphene control experiment, the active electrode was placed over Oz and the reference electrode was placed on the right cheek (see *Figure 4B*). In experiment 4, the active electrode was placed between F3 and Fz while the reference electrode was placed over the right cheek (see *Figure 5A*). In experiment 5, the electrodes were placed over Oz and between F3 and Fz.

In experiments 1, 2 and 4 each participant received both anodal and cathodal stimulations for a total of 30 min (15 min anodal, 15 min cathodal) in two separate experimental sessions separated by a washout period of at least 24 hr, the order of which was randomized and counterbalanced across participants.

In experiment 3, each participant received both anodal and cathodal stimulations for a total of 30 min (15 min anodal, 15 min cathodal) in two separate experimental sessions separated by a washout period of at least 24 hr, the order of which was randomized and counterbalanced across participants. These participants also received sham stimulation on one of the 3 days of testing; during the sham stimulation the machine ramped on for 5 s and switched off after 30 s of stimulation (ramping off over 5 s).

In experiments 5, each participant received both anodal-occipital + cathodal-prefrontal, and cathodal-occipital + anodal-prefrontal stimulations for a total of 30 min (15 min anodal, 15 min cathodal) in two separate experimental sessions separated by a washout period of at least 24 hr, the order of which was randomized and counterbalanced across participants. These participants also received sham stimulation on one of the 3 days of testing, during the sham stimulation the machine ramped on for 5 s and switched off after 30 s of stimulation (ramping off over 5 s).

The experimenter was not blind to which polarity condition the participant was in from day to day. In experiment 1 the intensity used for stimulation was 1 mA, for all other experiments 1.5 mA was used.

In the control tDCS modulation of phosphene thresholds experiment, the tDCS parameters were the same as in experiment 2. The intensity of the stimulation was set to 1.5mA and the active electrode placed over Oz and the reference electrode on the right cheek (see main text Materials and methods section and *Figure 4B*.)

## Statistical analysis of tDCS and TMS data

All correlational analysis, ANOVA's and t-tests were run in SPSS (IBM, Armonk), and the LME's were run in R (R Core Team, 2018) using the lme4 package (*Bates et al., 2015*).

To assess the effect of tDCS on imagery strength, we calculated the percent change in priming for each participant from baseline. This was then divided by their baseline imagery strength, to control for an individual's imagery strength, and multiplied by 100 to obtain a percent change score ((block (n) - (average of two tDCS blocks before stimulation)/ (average of two tDCS blocks before stimulation))*100).

These percent change scores were used as we were interested in how much each individual's imagery strength increased or decreased relative to their baseline imagery scores for the different tDCS polarities. Analyzing data in this way also normalized the data for individual differences in visual imagery strength.

For all linear mixed-effects models with cathodal + anodal stimulation conditions, tDCS polarity (cathodal and anodal), block (D1, D2, P1, P2 – see *Figure 4—figure supplement 1A* for timeline), and order of stimulation (cathodal stimulation first or second) were entered (without interaction terms) into the model as fixed effects. As random effects intercepts for subjects were entered into the model. p-Values were obtained by likelihood ratio tests of the full model with tDCS polarity included versus the model without tDCS included.

For all linear mixed-effects models with cathodal + anodal + sham stimulation conditions, tDCS polarity (cathodal, anodal, sham), block (D1, D2, P1, P2 – see *Figure 4—figure supplement 2A* for timeline), and order of stimulation (cathodal stimulation first, second or third) were entered (without interaction terms) into the model as fixed effects. As random effects intercepts for subjects were

entered into the model. p-Values were obtained by likelihood ratio tests of the full model with tDCS polarity included versus the model without tDCS included.

## Acknowledgements

We thank Wolf Singer for his support in the project and for his helpful comments on the manuscript. We would also like to thank Roger Koenig for helpful comments. JP is supported by Australian NHMRC grants APP1024800, APP1046198, and APP1085404 and a Career Development Fellowship APP1049596 and ARC discovery projects DP140101560 and DP160103299. An International Postgraduate Research Scholarship and a Brain Sciences UNSW PhD Top Up Scholarship supported JB. RK was supported by an Australian Postgraduate Award.

## Additional information

### Funding

| Funder | Grant reference number | Author |
| --- | --- | --- |
| National Health and Medical Research Council | APP1024800 | Joel Pearson |
| National Health and Medical Research Council | APP1046198 | Joel Pearson |
| National Health and Medical Research Council | APP1085404 | Joel Pearson |
| National Health and Medical Research Council | APP1049596 | Joel Pearson |
| Australian Research Council | DP140101560 | Joel Pearson |
| Australian Research Council | DP160103299 | Joel Pearson |
| University of New South Wales | International Postgraduate Research Scholarship | Johanna Bergmann |
| University of New South Wales | Brain Sciences UNSW PhD Top Up Scholarship | Johanna Bergmann |
| Australian Federal Government | Australian Postgraduate Award | Rebecca Keogh |

The funders had no role in study design, data collection and interpretation, or the decision to submit the work for publication.

### Author contributions

Rebecca Keogh, Johanna Bergmann, Conceptualization, Data curation, Formal analysis, Investigation, Visualization, Methodology; Joel Pearson, Conceptualization, Resources, Supervision, Funding acquisition, Project administration

### Author ORCIDs

Rebecca Keogh ![ORCID] https://orcid.org/0000-0003-4814-433X
Johanna Bergmann ![ORCID] https://orcid.org/0000-0003-1968-9680
Joel Pearson ![ORCID] https://orcid.org/0000-0003-3704-5037

### Ethics

Human subjects: All MRI research was carried out in Germany at the Max Planck Institute for Brain Research and all brain stimulation research (tDCS and TMS) was carried out in Australia at the University of New South Wales. Written informed consent was obtained from all participants and the ethics committee of the Max Planck Society approved the MRI study and the ethics committee of the University of New South Wales approved the tDCS and TMS studies (HC12030 & HC17031).

Decision letter and Author response
Decision letter https://doi.org/10.7554/eLife.50232.sa1
Author response https://doi.org/10.7554/eLife.50232.sa2

## Additional files

### Supplementary files
- Source code 1. LME code for r analysis.
- Supplementary file 1. fMRI tables.
- Transparent reporting form

### Data availability

All data analysed during this study are included in the manuscript and supporting files. Source data files are provided for Figures 3, 4 and 5 and Figure 2 - figure supplement 1.

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

## Appendix 1

### Additional mean fMRI intensity analysis

Our data are compatible with the hypothesis that the resting levels of early visual cortex activity are negatively related to imagery strength. However, mean fMRI intensity levels are not a commonly used measure. Individual variation in this parameter is influenced by many factors that are challenging to control for, for example differences in proton density (*Hashemi et al., 2010*). Additionally, fMRI activity at rest is influenced by a number of factors other than neural activity, such as non-neuronal physiological fluctuations or scanner noise (*Bergmann et al., 2016a*). We also cannot exclude the possibility that individual differences in vasculature influenced our results. Nonetheless, previous research has shown that the fMRI signal during resting state is strongly reflective of underlying neural activity (*Scholvinck et al., 2010*; *Bianciardi et al., 2009*).

### The influence of brain anatomy on the fMRI mean intensity relationships with behavior

By normalizing the individual fMRI signal levels of our ROIs using the whole brain's signal intensity, we aimed to control for some non-neuronal influences that may affect the individual brain in its entirety (e.g. scanner noise). We also excluded the possibility that differences in head size contributed to the relationship between the ROIs' mean fMRI signal levels and individual behavior. Using cortical surface area, thickness and volume, respectively, as proxies for head size, there was no significant relationships between these measures and either the behavioral or fMRI data (all p>0.09, see Supplementary Table S4). Accordingly, partialling out these factors did not change the pattern of significant results (all p<0.02). Interestingly, however, there were significant relationships between normalized V1 mean intensity and V1 anatomical measures, the latter of which have been reported to show a relationship with imagery in a previous study (*Bergmann et al., 2016a*). More specifically, normalized V1 mean intensity correlated positively with V1 surface size (r = 0.503, p=0.004), and negatively with cortical thickness (r = -.408, p=0.023). These relationships were non-significant for V2, V3 and lateraloccipital cortex surface size/thickness and fMRI mean activity - only for V2, there was a positive relationship between fMRI activity and volume (volume is a combined measure of surface and thickness): r = 0.369, p=0.041 (see Supplementary Table S5). Partialling out V1 surface size lowered the correlation between normalized V1 mean intensity and imagery slightly below significance level (r = -.32, p=0.085). The correlation remained significant when partialling out V1 thickness (r = -.449, p=0.013). For normalized V2 mean intensity, the correlation with imagery lowered below significance level when partialling out V2 volume (r = -.349, p=0.059). As the negative relationship between fMRI intensity and imagery still exists when anatomy is taken into consideration (although in the case of partialling out V1 surface size and V2 volume, the power is too low for statistical significance), we believe it is unlikely that the relationship can be completely explained by relations to V1 anatomy.

### The influence of head motion on the fMRI mean intensity relationships with behavior

To rule out the possibility that individual differences in head motion contributed to the relationship between imagery and individual intensity variability, we re-analysed the data in two ways: (1) when partialling out overall head movement, and (2) after excluding volumes with strong head movements ('scrubbing'; see Materials and methods and *Power et al., 2012*). Partialling out overall head movement again revealed significantly negative relationships between imagery strength and V1, V3 and lateral occipital area (V1: $r_{s\ xy.z} = -0.495$, p=0.025; V3: $r_{s\ xy.z} = -0.427$, p=0.03; lateral occipital: $r_{s\ xy.z} = -0.462$, p=0.025; FDR-adjusted p-values for multiple comparison correction). For V2, the relationships with

imagery was just below significance (V2: $r_{s\ xy.z}$ = −0.374, p=0.051; FDR-adjusted p-value). When computing the relationships after excluding volumes with strong head movement, a similar pattern of relationships emerged: the relationship of imagery strength with V1 and lateral occipital area was significantly negative (V1: r = −0.442, p=0.033; lateral occipital: r = −0.489, p=0.025). For V2 and V3, the negative relationship was below significance (V2: r = −0.318, p=0.081, V3: r = −0.331, p=0.081; all p-values FDR-adjusted). Based on these results, we deem it unlikely that individual head motion had a major impact on the observed relationships.

## Mock difference scores tDCS

### 1 mA Occipital Stimulation

There was no significant main effect of tDCS polarity for the percentage change in mock priming (F(1,15) = 2.91, p=0.11) or block on mock priming (Greenhouse-Geisser correction for violation of sphericity, F(1.90,28.51) = 1.69, p=0.20), and no interaction between the two (Greenhouse-Geisser correction for violation of sphericity, F(1.68, 25.22)=0.06, p=0.92). When averaging across all of the blocks neither the cathodal or anodal condition mock priming was significantly different to 0 (t(15) 1.81, p=0.10 and t(15) −0.10, p=0.92, respectively).

### 1.5 Occipital Stimulation

There was no significant main effect of tDCS polarity for the percentage change in mock priming (F(1,15) = .07, p=0.80) or block on mock priming (Greenhouse-Geisser correction for violation of sphericity, F(1.66, 24.97)=1.23, p=0.30), and no interaction between the two (Greenhouse-Geisser correction for violation of sphericity, F(1.59, 23.87)=1.03, p=0.36). When averaging across all of the blocks neither the cathodal or anodal condition mock priming was significantly different to 0 (t(15) 1.02, p=0.33 and t(15). 66, p=0.52, respectively).

### 1.5 Occipital stimulation (including sham)

There was no significant effect of tDCS polarity for the percentage change in mock priming (Mixed-effects analysis due to attrition Greenhouse-Geisser correction for violation of sphericity,: F(1.99, 28.82)=1.23, p=0.31) or block on mock priming (Mixed-effects analysis due to attrition Greenhouse-Geisser correction for violation of sphericity: F(2.12, 31.80)=0.60, p=0.56) and no interaction between the two (Mixed-effects analysis due to attrition Greenhouse-Geisser correction for violation of sphericity: F(3.40, 46.49)=0.28, p=0.87). When averaging across all of the blocks none of the polarity conditions were significantly different to 0 (sham: t(13) 1.80, p=0.09, cathodal: t(15). 16, p=0.87, and anodal: t(15) = .70, p=0.50).

### 1.5 Prefrontal stimulation

There was no main effect of tDCS polarity for the percentage change in mock priming (F(1,15) = .02, p=0.89) or block on mock priming (Greenhouse-Geisser correction for violation of sphericity, F(1.81, 27.20)=1.18, p=0.32), and no interaction between the two (Greenhouse-Geisser correction for violation of sphericity, F(1.78, 26.73)=1.73, p=0.20). When averaging across all of the blocks neither the cathodal or anodal condition mock priming was significantly different to 0 (t(15) −0.18, p=0.86 and t(15) −0.59, p=0.56, respectively).

### 1.5 Prefrontal + occipital stimulation (including sham)

There was no significant effect of tDCS polarity for the percentage change in mock priming (Mixed-effects analysis due to attrition Greenhouse-Geisser correction for violation of sphericity,: F(1.06, 19.08)=0.53, p=0.76) or block on mock priming (Mixed-effects analysis due to attrition Greenhouse-Geisser correction for violation of sphericity: F(1.812, 32.62)=0.60,

p=0.42) and no interaction between the two (Mixed-effects analysis due to attrition Greenhouse-Geisser correction for violation of sphericity: F(2.75, 42.11)=0.46, p=0.56). When averaging across all of the blocks none of the polarity conditions were significantly different to 0 (sham: t(17). 59, p=0.55, cathodal occipital + anodal pFC: t(15). 23, p=0.82, and anodal occipital + cathodal pFC: t(15) = .58, p=0.57).

