## [Decision Letter]

**Acceptance summary:**

The ability to create vivid mental images varies considerably across people. This study examines neural correlates of this variability. The authors show that stronger imagery is associated with both lower resting activity (neuroimaging) and lower excitability (phosphene thresholds) in early visual cortex. Supporting these correlational data, causal experiments show that decreasing cortical excitability using transcranial direct current stimulation increases imagery strength. Together these results suggest potential neural mechanisms that determine the strength of mental images in different people.

**Decision letter after peer review:**

Thank you for submitting your article "Cortical Excitability controls the strength of mental imagery" for consideration by *eLife*. Your article has been reviewed by three peer reviewers, and the evaluation has been overseen by a Reviewing Editor and Floris de Lange as the Senior Editor. The following individual involved in review of your submission has agreed to reveal their identity: Nadine Dijkstra (Reviewer #3).

The reviewers have discussed the reviews with one another and the Reviewing Editor has drafted this decision to help you prepare a revised submission.

Summary:

This manuscript by Keogh et al. investigates the relationship between the excitability of early visual cortex and the strength of visual imagery. They find a negative relationship, such that decreased cortical excitability is related to increased imagery strength. Convergent results come from three different techniques, which reviewers found very impressive. Reviewers further found the manuscript to be well written, and that the reporting of the methods and results is relatively thorough and transparent. Individual reviewers also had questions about the assumptions underlying the methodology and felt that the claims about novelty and specificity need to be better supported or toned down in a revised version.

Essential revisions:

1) The authors make relatively strong statements regarding the specificity of the observed relationship between excitability and imagery. But the specificity of the effects for imagery has not been demonstrated. Without additional control tasks that can disentangle imagery from other more basic perceptual processes, the current conclusions are too strong. For instance, to really argue that "Here we show the first evidence that pre-existing levels of neural excitability and spontaneous resting activity in visual cortex can influence the strength of mental representations," more controls are required. In particular, the authors should add results from a control experiment demonstrating that similar effects are not observed for perceptual performance. Alternatively, they need to tone down their conclusions and specificity claims throughout the manuscript.

2) The authors make strong claims regarding novelty. A pubmed search indicates that there are actually quite a few studies assessing links between excitability of visual cortex and mental image strength (e.g. luminance), yet many are not cited. There is also much evidence from brain stimulation studies linking imagery to cortical excitability and those are not considered. Overall, it appears that the authors are being somewhat unfair and perhaps inflating the novelty of their findings.

3) Is normalized fMRI signal intensity a reliable marker of cortical excitability? Which neural processes are captured by the normalized resting state BOLD measure? How is this measure related to potential confounds, such as vasculature, distance to head coil, and anatomical features (cortical thickness)? The authors should provide more evidence for the validity of this method.

4) Similarly, what is the evidence for the relationship between phosphene thresholds and cortical excitability? The authors state that "The magnetic strength needed to induce a phosphene is a reliable and non-invasive method to measure cortical excitability." This needs to be discussed in more detail. Can they provide any references to back this up? This seems important, because while this seems very intuitive, alternative neural mechanisms for phosphene threshold can be imagined. In addition, many things can potentially modulate phosphene thresholds, including coil-skull distance, existing brain state etc. Finally, is there previous evidence that fMRI signal intensity is related to cortical excitability as measured by TMS? If yes, please provide more details and references. If no, explain what the differences between these two measures might be and why it is interesting to look at both.

5) Another set of issues regards the structure of the manuscript and the theoretical motivation of the studies. The manuscript has a very descriptive character, summing up the results of a set of different experiments. Greater attempts should be made to provide a mechanistic explanation of the results that will increase our understanding of individual differences in mental imagery strength. For example, the motivation behind the link between imagery strength and cortical excitability at rest seems very exploratory. It will help if the authors better explain what cortical excitability means exactly and how this relates to other brain measures that have been linked to imagery strength. Also, it is not entirely clear why the relationship between resting state activity and imagery strength should disappear in the presence of background luminance. You could interpret this as a measure of how well participants can inhibit bottom-up input during mental imagery, which might therefore show an even stronger negative relation to visual cortex activity. Also, the notion of background neural noise is interesting but should be explained better. If this is a measure of SNR in the visual cortex, it should have more general effects than just on imagery. For example, you would expect perceptual processing to also be negatively related to neural noise. In contrast, the authors state in the Discussion that perceptual sensitivity is positively related to visual cortex excitability (but they do not provide any references for this statement). Finally, the Discussion focuses solely on the visual cortex results whereas the frontal cortex results are equally compelling. More effort should be made into reworking the results of the different experiments into a coherent narrative.

6) It would be important to discuss the contrast between the current findings and earlier studies that found a positive relation between BOLD in visual cortex and imagery strength and that showed cross-decoding between perceived and imagined objects. This deserves more attention. The authors suggest that this discrepancy is due to other studies subtracting baseline BOLD activation to estimate their effects. However, the authors also find an 'online' effect of low excitability and imagery strength in the tDCS experiments. The authors should try to explain this discrepancy better, perhaps elaborating on this notion of neural noise and how it reflects different brain measurements in different ways.

7) Please provide source data for all figures.

[Editors' note: further revisions were suggested prior to acceptance, as described below.]

Thank you for resubmitting your work entitled "Cortical Excitability controls the strength of mental imagery" for further consideration by *eLife*. Your revised article has been evaluated by Floris de Lange as the Senior Editor, a Reviewing Editor and two peer reviewers.

The manuscript has been improved but there are some remaining issues that need to be addressed before acceptance, as outlined below:

1) Rebuttal to point 3: "it is difficult to interpret how V1 surface size/thickness alone should present a confound to the observed relationship between mean fMRI activity and behavior." I'm not sure I agree; one of the authors, Bergmann, has published many findings relating anatomical features of V1, especially surface size, to vividness and precision of imagery. So given the relationship between normalised mean activity and V1 surface size and thickness, couldn't these anatomical features explain the current findings, rather than normalised mean activity, which is a measure we don't really know the physiological basis of? At the very least these relationships between mean fMRI signal and V1 anatomy should be included in the manuscript, since the current statement that "Using cortical surface area, thickness and volume, respectively, as proxies for head size, there was no significant relationships between these measures and either the behavioral or fMRI data (all p >.09)" seems somewhat misleading.

2) The results of the normalized fMRI measure are a bit unclear and take away from the strengths of the paper. Firstly, the authors admit that they are not sure what exactly reflects in terms of neural processing. Secondly, they find a significant relationship between this measure and V1 surface size as well as thickness, which they have previously reported shows a correlation with imagery strength by itself (Bergmann et al., 2016). This means that at least part of these findings might be explained via that mechanism and do not by themselves contribute anything new. Because of this, I think it would improve the readability and clarity of the manuscript if this section on normalized fMRI activity is shortened and treated as an exploratory analysis to identity the sites of brain stimulation. It might then also make sense to only show the whole-brain analysis in the main paper and add the ROI analyses as supplementary figures.

---

## [Author Response]

Essential revisions:1) The authors make relatively strong statements regarding the specificity of the observed relationship between excitability and imagery. But the specificity of the effects for imagery has not been demonstrated. Without additional control tasks that can disentangle imagery from other more basic perceptual processes, the current conclusions are too strong. For instance, to really argue that "Here we show the first evidence that pre-existing levels of neural excitability and spontaneous resting activity in visual cortex can influence the strength of mental representations," more controls are required. In particular, the authors should add results from a control experiment demonstrating that similar effects are not observed for perceptual performance. Alternatively, they need to tone down their conclusions and specificity claims throughout the manuscript.

We thank the reviewers for their suggestion. We have added a section in the Discussion about the limitations to our study in relation to potential other cognitions involved in our task, which we hope provides more balance and sheds light on the current limitations to our findings (Discussion, second paragraph):

“It should be noted that while previous work has demonstrated that this measure of imagery strength can be separated from feature-based attention [Laakso et al., 2019], we cannot explicitly rule out the possibility that tDCS was improving other cognitive mechanisms that are involved in this task such as sustained or selective attention. […] It may be the case that all forms of visual imagery are improved by prefrontal cortex stimulation, due to an increase in the strength of top-down signals, whereas only images that require activation of low-level sensory features, such as color or orientation, will be influenced by the excitability of early visual cortex.”

2) The authors make strong claims regarding novelty. A pubmed search indicates that there are actually quite a few studies assessing links between excitability of visual cortex and mental image strength (e.g. luminance), yet many are not cited. There is also much evidence from brain stimulation studies linking imagery to cortical excitability and those are not considered. Overall, it appears that the authors are being somewhat unfair and perhaps inflating the novelty of their findings.

There are indeed some other visual imagery and brain stimulation studies that exist, some of which we have referenced in our Discussion. For example:

“Additionally some TMS studies have found that during visual imagery, visual cortex excitability increases (Sparing et al., 2002).”

There is also a study assessing rTMS and visual imagery which we reference in our Discussion:

“For example, one study found that applying 1Hz of TMS to area BA 17 (primary visual cortex), slowed responses in a task where individuals had to imagine stripes (or were perceptually shown stripes) and answer questions about these images (Kosslyn, 1999).”

There is another study by Cattaneo et al., 2011 that is very similar to Sparing et al., 2002, which we did not reference: which the reviewer is referring too with “e.g. luminance” (Cattaneo et al., 2011). This study assessed phosphene thresholds while participants simultaneously remember/imagine visual images and found that phosphene thresholds decrease/cortical excitability increases when imaging/remembering images (similar to Sparing et al., 2002). There is some evidence from this study that the luminance of the remembered/imagined image modulates phosphene thresholds in different ways – although this evidence varies as a function of TMS machine output and is somewhat mixed, making it hard to interpret.

It is important to note that these studies show that while imagining (online task performance) visual cortex excitability increases and this suggests that visual imagery uses the visual cortex. These studies were instrumental in providing evidence for the involvement of early visual cortex during visual imagery, helping to resolve the imagery debate. However, none of these studies assess individual differences in visual imagery strength, and how the resting levels of excitability in visual cortex might influence the strength with which one can imagine. This is what we do in the current study, and what the claim of novelty refers to. We have now added the Cattaneo et al., 2011 reference to the discussion on previous brain stimulation results and have also added a paragraph to the Introduction with more references to previous work on both the neuroimaging and brain stimulation work that exists assessing the relationship between perception and imagery:

“To date, much of the research in the field of visual imagery has focused on the similarities between visual imagery and perception, due to a long-ranging debate around whether visual imagery can be depictive and/or pictorial, referred to as the “imagery debate” [Pearson and Kosslyn, 2015]. […] Brain stimulation research has similarly investigated whether the early visual cortex is involved during visual imagery with findings demonstrating that, like motor imagery, visual cortex excitability increases during imagery [Cattaneo et al., 2011; Sparing et al., 2002].”

3) Is normalized fMRI signal intensity a reliable marker of cortical excitability? Which neural processes are captured by the normalized resting state BOLD measure? How is this measure related to potential confounds, such as vasculature, distance to head coil, and anatomical features (cortical thickness)? The authors should provide more evidence for the validity of this method.

We acknowledge that the normalized fMRI signal measure we use is not common. However, the relationship we observe holds across the many various control analyses we have run, and the same pattern appears in two different resting-state measurements. This leads us to think that the observed relationships might not just be coincidental and are too consistent to be ignored. However, we are aware that these data should be interpreted with caution and not by themselves (and this is the reason why we included the TMS and tDCS experiments).

With regards to the neural processes captured, we are not aware of any studies assessing individual resting fMRI signal intensity with neurotransmitters. There is, however, evidence that the excitatory transmitter Glutamate and fMRI BOLD are related (e.g. Ip et al., 2016 and 2019, Boillat et al., 2019), such that during tasks (online) increases in Glutamate in the visual cortex are accompanied by an increase in BOLD signal, and a decrease in Glutamate is accompanied by a decrease in BOLD signal.

Although we find it hard to think of reasons why and how individual differences in vasculature would affect our findings, we cannot exclude this possibility – this would require additional measurements like VASO or ASL, which we don’t have with these data. We have added this issue as another additional potential confound to the manuscript. Regarding the distance to head coil: since the occipital cortex is located posteriorly, and participants lay in supine position, the inter-subject difference in (posterior) distance to the head coil should be negligible, leading us to assume that it is unlikely that differences in the distance to the head coil played a role. As a proxy for head size, which would be an indicator for the distance to the head coil (at least the distance to the forehead, which would vary in contrast to the backside of the head), we partialled out gray matter volume (or surface, or thickness); this did not change the found relationships. There is also no significant relationship between normalized mean fMRI activity (of V1, V2, V3, and lateraloccipital cortex) and whole brain surface size, thickness, or volume (all *p* >.09). We have added whole-brain thickness analyses to the manuscript (the previous version only contained whole brain surface size and volume, which had no significant relationship, either; all *p* >.16).

Interestingly, when only looking at the relationship between mean fMRI activity and anatomical measures of the ROIs (instead of anatomical measures of the whole brain), we do indeed find a significant relationship between V1 mean fMRI activity and (functionally defined, central) V1 surface size (positive, r=.503, p=.004), as well as with V1 thickness (negative, r=-.408, p=.023). However, these relationships are non-significant for V2, V3 and lateraloccipital cortex surface size/ thickness and fMRI mean activity – only for V2, there is a positive relationship between fMRI activity and volume (volume is a combined measure of surface and thickness): r=.437, p=.014. In light of the fact that fMRI mean activity does not seem to be related to overall head size (as proxied by cortical volume, surface, and thickness), it is difficult to interpret how V1 surface size/thickness alone should present a confound to the observed relationship between mean fMRI activity and behavior.

4) Similarly, what is the evidence for the relationship between phosphene thresholds and cortical excitability? The authors state that "The magnetic strength needed to induce a phosphene is a reliable and non-invasive method to measure cortical excitability." This needs to be discussed in more detail. Can they provide any references to back this up? This seems important, because while this seems very intuitive, alternative neural mechanisms for phosphene threshold can be imagined.

There are many studies that have used phosphene thresholds as a proxy for cortical excitability, for a non-exhaustive list see: (Bestmann et al., 2007; Cowey and Walsh, 2000; Romei et al., 2008; Stewart, Walsh and Rothwell, 2001). The use of phosphenes as an indicator of cortical excitability is based on motor thresholds, which are used to measure the excitability of motor cortex. Motor thresholds can be measured using MEP’s (motor evoked potentials) or they can be measured through observing magnetically elicited movement in a limb (typically the thumb), which is thought to be comparable to the elicitation of phosphenes.

Much of the research into visual cortex excitability and phosphene thresholds have occurred in populations who have abnormal/enhanced cortical excitability, such as epilepsy and migraines (Afra et al., 1998; Aurora et al., 1998; Aurora and Welch, 1998; Brigo et al., 2013; Young et al., 2004). Studies using drugs that reduce cortical excitability have also shown that phosphene thresholds are increased (decreased cortical excitability) after the administration of these drugs. For example, phosphene thresholds have been demonstrated to increase (= reduce cortical excitability) as compared to baseline/pre-treatment in migraine patients who are taking the GABA agonist drug Valproate (Mulleners et al., 2002; Palermo et al., 2009).

In addition, there is some evidence that phosphene thresholds are increased in migraineurs who take the drug Topiramate (multiple modes of action as GABA agonist as well as being an NMDA antagonist, (Artemenko et al., 2008)). Recent research has shown that phosphene perception is related to glutamate levels in the early visual cortex, with higher levels of glutamate correlating negatively with phosphene thresholds, which demonstrates that those with higher proportions of glutamate (an excitatory neurotransmitter) have more excitable visual cortices when measured using phosphene thresholds (Terhune et al., 2015). Taken together, these results provide evidence that phosphene thresholds do measure cortical excitability, although whether they measure only inhibition, excitation or a combination of the two is still not fully known.

We thank the reviewers for bringing up this point, as a reader who is not familiar with non-invasive brain stimulation may not have this background knowledge. A section on measuring and manipulating cortical excitability has now been added to the Materials and methods. We hope that the paper is now accessible to a more general audience with the inclusion of the following section (Materials and methods):

“Measuring and Manipulating Cortical Excitability with Brain Stimulation (TMS and tDCS): Cortical excitability can be broadly defined as the ease with which a neuron, or a population of neurons, can produce an action potential. […] Taken together these results provide evidence that phosphene thresholds are a good proxy for measuring cortical excitability, although whether they measure inhibition, excitation or a combination of the two is still not fully known.”

In addition, many things can potentially modulate phosphene thresholds, including coil-skull distance, existing brain state etc.

There is indeed research suggesting that brain state can affect phosphene thresholds, such that focusing on a spatial location can decrease phosphene thresholds (Bestmann et al., 2007). For this reason, all participants were instructed to maintain fixation on a central location when the phosphene thresholds were being tested (see Materials and methods section):

“The participants were instructed to relax and stare forward at a fixation dot in the middle of the black cardboard and to let the experimenter know if they saw any sort of visual disturbances on the cardboard and if they did see something to describe what they saw and in which quadrant they saw it.”

Finally, is there previous evidence that fMRI signal intensity is related to cortical excitability as measured by TMS? If yes, please provide more details and references. If no, explain what the differences between these two measures might be and why it is interesting to look at both.

This is an interesting question. To the best of our knowledge, there have been no studies to date to assess TMS induced phosphene thresholds and fMRI signal intensity. Nonetheless, previous work has shown – separately – that both BOLD and phosphene thresholds are related to glutamate concentrations. One of the main reasons for assessing both MRI and phosphene thresholds, is that (as mentioned in the response to comment #3), fMRI signal intensity is not a frequently used measure, and as such we wanted to assess whether similar correlations with imagery existed with a more accepted measure of cortical excitability (TMS induced phosphene thresholds).

5) Another set of issues regards the structure of the manuscript and the theoretical motivation of the studies. The manuscript has a very descriptive character, summing up the results of a set of different experiments. Greater attempts should be made to provide a mechanistic explanation of the results that will increase our understanding of individual differences in mental imagery strength. For example, the motivation behind the link between imagery strength and cortical excitability at rest seems very exploratory. It will help if the authors better explain what cortical excitability means exactly and how this relates to other brain measures that have been linked to imagery strength.

We thank the reviewers for this suggestion. We have now added more background about the motivation for assessing visual cortical excitability and imagery (see below), as well as a section defining cortical excitability and the measures of cortical excitability we use in this study (also see above response to comment #4):

“Taken together, these studies suggest that visual cortex is linked to the subjective vividness of visual imagery. […] It is known that the excitability of visual cortex varies substantially across individuals, and as such may be a candidate for driving some of the observed interindividual differences in visual imagery strength.”

Also, it is not entirely clear why the relationship between resting state activity and imagery strength should disappear in the presence of background luminance. You could interpret this as a measure of how well participants can inhibit bottom-up input during mental imagery, which might therefore show an even stronger negative relation to visual cortex activity.

The point the reviewers raise is a fair one. Our notion is that imagery ability isn’t necessarily linked to the ability to *inhibit* bottom-up input – rather, we believe that the top-down/internally-generated imagery signals might be stronger in individuals with strong imagery (against the background ‘noise’/bottom-up input). Increased bottom-up stimulation (=’noise’ in the form of afferent visual stimuli) would then lead to a lower signal-to-noise ratio for the top-down imagery signal, reducing imagery strength.

The inclusion of the luminance background was aiming mainly to assess whether resting-state activity correlated with the voluntary control of binocular rivalry, rather than the content/strength of visual imagery. Our reasoning behind this is that irrelevant visual information (in the form of background luminance) is thought to interfere with low-level sensory creation of visual images in mind, but as it is never co-presented with the binocular rivalry display it should not interfere with an individual’s voluntary control of visual imagery. As such if our results were purely driven by voluntary control of binocular rivalry, rather than the strength of an individual’s visual imagery, the observed correlations should still exist with the luminous background.

Nonetheless, even though we favour the notion depicted above, as the reviewer points out, it is not possible to exclude the possibility that stronger imagery is not only related to a *stronger top-down* imagery signal, but also to a better ability to *inhibit* bottom-up input. In this case, the imagery-under-luminance test would be influenced by both of these factors, making the results from the test hard to interpret. Given the reviewer’s comment, and since we cannot disentangle this with the given data set, this point requires more attention than is possible in the present study and with the extensive length of the manuscript, we have decided to remove this section.

Also, the notion of background neural noise is interesting but should be explained better. If this is a measure of SNR in the visual cortex, it should have more general effects than just on imagery. For example, you would expect perceptual processing to also be negatively related to neural noise. In contrast, the authors state in the Discussion that perceptual sensitivity is positively related to visual cortex excitability (but they do not provide any references for this statement). Finally, the Discussion focuses solely on the visual cortex results whereas the frontal cortex results are equally compelling. More effort should be made into reworking the results of the different experiments into a coherent narrative.

It is indeed an intriguing question as to why and how spontaneous background activity may affect perceptual and imagery processes differentially. Very basic detection type perceptual tasks tend to find that high levels of spontaneous cortical activity/excitability lead to better performance (for examples see: (Antal, Nitsche and Paulus, 2001;Behrens et al., 2017; Ding et al., 2016; Kraft et al., 2010; Scholvinck, Friston and Rees, 2012). This interpretation fits with signal-to-noise ratio theories: It may be the case that in very simple detection-only tasks, increasing excitability leads to improved performance because there are no competing responses.

In contrast to stimulus detection, the relationship between stimulus discrimination and cortical excitability seems more complex, with some studies finding heightened cortical excitability improves performance, while others have found reducing cortical excitability improves performance (Antal, Nitsche, et al., 2004; Reinhart et al., 2016; Waterston and Pack, 2010). This suggests that in some cases perceptual ability, like imagery, can be improved when spontaneous activity is reduced. However, this may be limited to tasks that require a ‘sharp’ representation of a specific stimulus. It may be the case that when more than pure detection is required, increasing cortical excitability may not just increase the target stimuli activity, but also all other non-target related neural activity. With an overall lower spontaneous background activity, the stimulus-specific signal may stand out more, making it easier to distinguish it from other stimuli. With regards to imagery-related signals, which are presumably relatively feeble, reducing background ‘noise’ should have a similar effect, such that the signal of the imagined stimulus becomes discernible against other competing signals (which may be perceptual etc).

Another factor to consider is the possibility that tDCS stimulation has a tendency to *selectively* reduce non-imagery related signals and background noise. Given the placement of the tDCS pads on the skull, tDCS appears to have a higher impact on the more superficial cortical layers than on the deeper cortical layers, as the superficial layers are closer to the current source (Komarov et al., 2019). Interestingly, research from one of the authors of our study indicates that imagery-related signals are only decodable from deep cortical layers (Bergmann, Morgan, and Muckli, 2019). As a consequence, tDCS may attenuate signals in the mid- and superficial layers more than those in the deep layers, thereby causing a relative advantage of deep-layer imagery-related signals over the ones arising in the other layers.

We have now added references to these perception and cortical excitability studies in the manuscript. In addition, we have added more discussion regarding the potential relationship between signal-to-noise ratios and cortical excitability, and the theoretical role it plays in visual imagery strength. We have also added a section discussing how both visual cortex and pre-frontal cortex might work in concert to alter the signal-to-noise ratio in visual imagery generation and maintenance (Discussion section, third paragraph onwards):

“Over the last 30 years, empirical work has demonstrated many commonalities between imagery and visual perception (see [Pearson et al., 2015, Dijkstra, Bosch and van Gerven, 2019] for a review). […] Increased top-down signals might also allow for a greater inhibition of non-signal related neural noise further down the cortical hierarchy, resulting in a higher signal-to-noise ratio in the visual cortex.”

6) It would be important to discuss the contrast between the current findings and earlier studies that found a positive relation between BOLD in visual cortex and imagery strength and that showed cross-decoding between perceived and imagined objects. This deserves more attention. The authors suggest that this discrepancy is due to other studies subtracting baseline BOLD activation to estimate their effects. However, the authors also find an 'online' effect of low excitability and imagery strength in the tDCS experiments. The authors should try to explain this discrepancy better, perhaps elaborating on this notion of neural noise and how it reflects different brain measurements in different ways.

Thank you for raising these points; these are indeed intriguing questions. We do find an online effect as well, and on reflection, the explanation we provided in our Discussion did not fully make the case we intended. We have re-written our Discussion to make it clearer that it is not necessarily the ‘onlineness’ of the task that is resulting in these differences, rather what is being measured by the tasks. These fMRI studies generally measure changes in BOLD while an individual is performing a task – it is possible that if we had used fMRI during our imagery task there may have been increased BOLD activity in visual cortex. What we mean to convey is that this increase in BOLD may be larger in those who have stronger visual imagery due to a lower level of resting activity. Our response to #5. also relates to some of the points raised here with regards to the potential mechanisms underlying the observed ‘online’ effect of low excitability and imagery strength in tDCS experiments. We have addressed this point in two ways. Firstly, we have added more background on previous fMRI studies and visual imagery into the Introduction:

“To date, much of the research in the field of visual imagery has focused on the similarities between visual imagery and perception, due to a long-ranging debate around whether visual imagery can be depictive and/or pictorial, referred to as the “imagery debate” [Pearson and Kosslyn, 2015]. […] A recent study found that trial-by-trial differences in imagery vividness were also related to the similarity of BOLD responses between imagery and perception [Dijkstra, Bosch and van Gerven, 2017].”

We have also re-written the Discussion point on ‘online’ BOLD experiments:

“Previous work has also found positive correlations between BOLD activity in the visual cortex and the vividness of visual imagery questionnaire [Cui et al., 2007; Ip et al., 2017]. […] It may be that participants with initially low visual cortex excitability are able to increase visual cortex activity more-so than those with higher-levels, and this could potentially explain the larger BOLD changes for individuals with stronger visual imagery.”

7) Please provide source data for all figures.

These have now been uploaded with the submission.

References:

Behrens, J. R., Kraft, A., Irlbacher, K., Gerhardt, H., Olma, M. C., & Brandt, S. A. (2017). Long-Lasting Enhancement of Visual Perception with Repetitive Noninvasive Transcranial Direct Current Stimulation. Front Cell Neurosci, 11, 238. doi:10.3389/fncel.2017.00238

Scholvinck, M. L., Friston, K. J., & Rees, G. (2012). The influence of spontaneous activity on stimulus processing in primary visual cortex. Neuroimage, 59(3), 2700-2708. doi:10.1016/j.neuroimage.2011.10.066

[Editors' note: further revisions were suggested prior to acceptance, as described below.]

The manuscript has been improved but there are some remaining issues that need to be addressed before acceptance, as outlined below:1) Rebuttal to point 3: "it is difficult to interpret how V1 surface size/thickness alone should present a confound to the observed relationship between mean fMRI activity and behavior." I'm not sure I agree; one of the authors, Bergmann, has published many findings relating anatomical features of V1, especially surface size, to vividness and precision of imagery. So given the relationship between normalised mean activity and V1 surface size and thickness, couldn't these anatomical features explain the current findings, rather than normalised mean activity, which is a measure we don't really know the physiological basis of? At the very least these relationships between mean fMRI signal and V1 anatomy should be included in the manuscript, since the current statement that "Using cortical surface area, thickness and volume, respectively, as proxies for head size, there was no significant relationships between these measures and either the behavioral or fMRI data (all p >.09)" seems somewhat misleading.

We thank the reviewers for this suggestion regarding including all of the relationships between fMRI mean intensity and V1 anatomy – all of the correlations between visual cortex ROIs’ anatomy and fMRI signal have been added as a table in Supplementary file 1 – see Supplementary table 5. (Please note one glitch that we spotted in the previous rebuttal letter. The relationship between V2 volume and V2 mean normalized fMRI intensity is r=.369, p=.041, *not* p=.437, p=.014 as stated in the previous rebuttal letter; the latter value erroneously referred to the correlation between V1 volume and V2 mean normalized fMRI intensity.) We apologize for the potentially misleading statement "Using cortical surface area, thickness and volume, respectively, as proxies for head size, there was no significant relationships between these measures and either the behavioral or fMRI data (all p >.09)", which refers to (overall) cortical anatomy (and not V1 anatomy). We have clarified this section now to avoid any misunderstanding and these correlational values have also been added as a new table (see Supplementary table 4 in Supplementary file 1).

With regards to the reviewers’ comment that the link between normalized mean intensity and imagery could simply be explained by their relationships to V1 anatomy, we do find some significant correlations between anatomy and mean fMRI intensity (see new Supplementary table 5 in Supplementary file 1). As it stands, it is difficult to tell exactly what the relationship between V1 anatomy and V1 intensity is. One interpretation could be that anatomy is the underlying causal factor of both imagery strength and normalized mean intensity. However, the relationships could also have a different causal structure. For example, normalized mean intensity could act as a mediator in the relationship between imagery and anatomy, or the reverse could be the case (anatomy could act as mediator). Further, it is also possible that another, independent factor could influence both anatomy and normalized mean intensity (e.g. during development), thereby leading to the association between the two.

When V1 surface size is partialled out, the correlation between mean normalized V1 intensity and imagery is lowered and moves slightly below significance level (r=-.32, p=.085). When partialling out V1 thickness, the relationship remains significant (r=-.45, p=.013). Seeing as the negative relationship between fMRI intensity and imagery still exists when anatomy is taken into consideration (although in the case of partialling out V1 surface size, the power is too low for statistical significance) we do not think this relationship can be fully ‘explained’ by relations to V1 anatomy. Rather, we think it likely that fMRI mean intensity holds its own separate relationship with imagery strength, but, as previously stated, with the present dataset this is difficult to disentangle. For this reason, and in line with a different suggestion from reviewers, the ROI analysis’ have been moved to the subsection “Additional mean fMRI intensity analysis”, with the cluster fMRI analysis being moved to become a main figure (see next point for further detail on this).

2) The results of the normalized fMRI measure are a bit unclear and take away from the strengths of the paper. Firstly, the authors admit that they are not sure what exactly reflects in terms of neural processing. Secondly, they find a significant relationship between this measure and V1 surface size as well as thickness, which they have previously reported shows a correlation with imagery strength by itself (Bergmann et al., 2016). This means that at least part of these findings might be explained via that mechanism and do not by themselves contribute anything new. Because of this, I think it would improve the readability and clarity of the manuscript if this section on normalized fMRI activity is shortened and treated as an exploratory analysis to identity the sites of brain stimulation. It might then also make sense to only show the whole-brain analysis in the main paper and add the ROI analyses as supplementary figures.

We thank the reviewers for this suggestion. We have now moved the scatterplots for the ROI analysis to supplementary figures (Figure 2—figure supplement 1) and have now used the cluster whole-brain analysis to create a new main text figure – Figure 2. We have also re-worded this fMRI data to emphasise its exploratory nature, see subsection “Visual Cortex and Visual Imagery Strength”.

We have also moved extra discussion on the limitations of fMRI intensity to the subsection “Additional mean fMRI intensity analysis”, along with the motion and anatomy analysis.

In addition to the suggested changes from the reviewers, we have made the binocular rivalry figure a stand-alone figure (Figure 1) and the TMS phosphene thresholds and mock data are their own stand-alone figure (Figure 3), to help the flow for readers.